# Bias-free driven ion assisted photoelectrochemical system for sustainable wastewater treatment

Qi Dang [1,6], Wei Zhang [2,6], Jiqing Liu[1], Liting Wang[1], Deli Wu[3], Dejin Wang[4], Zhendong Lei [3,5] ✉ & Liang Tang [1,4] ✉

Photoelectrochemical (PEC) systems have emerged as a prominent renewable energy-based technology for wastewater treatment, offering sustainable advantages such as eliminating dependence on fossil fuels or grid electricity compared to traditional electrochemical treatment methods. However, previous PEC systems often overlook the potential of ions present in wastewater as an alternative to externally applied bias voltage for enhancing carrier separation efficiency. Here we report a bias-free driven ion assisted photoelectrochemical (IAPEC) system by integration of an electron-ion acceptor cathode, which leverages its fast ion-electron coupling capability to significantly enhance the separation of electrons and holes at the photoanode. We demonstrate that Prussian blue analogues (PBAs) can serve as robust and reversible electron-ion acceptors that provide reaction sites for photoelectron coupling cations, thus driving the hole oxidation to produce strong oxidant free radicals at photoanode. Our IAPEC system exhibits superior degradation performance in wastewater containing chloride medium. This indicates that, in addition to the cations (e.g., $Na^+$) accelerating the electron transfer rate, the presence of $Cl^-$ ions further enhance efficient and sustainable wastewater treatment. This work highlights the potential of utilizing abundant sodium chloride in seawater as a cost-effective additive for wastewater treatment, offering crucial insights into the use of local materials for effective, low-carbon, and sustainable treatment processes.

The contamination of water by diverse refractory organic pollutants has emerged as a critical global concern, constituting a severe hazard to the environment and human health[1,2]. To address this issue, PEC technology, which combines the merits of both electrocatalysis and photocatalysis, has emerged as one of a promising and environmentally benign approaches for pollutant degradation[3–5].

The efficient separation of photogenerated charges plays a pivotal role in determining the performance and overall effectiveness of PEC systems[6,7]. Several strategies, including heterostructure construction, nanostructure engineering, and cocatalyst modification, have been employed to enhance the charge separation efficiency and improve the PEC performance of photoelectrodes[8,9]. However, the conventional PEC process necessitates the application of a bias voltage

[1]Key Laboratory of Organic Compound Pollution Control Engineering (MOE), School of Environmental and Chemical Engineering, Shanghai University, 200444 Shanghai, China. [2]Department of Chemistry, IRIS Adlershof & The Center for the Science of Materials Berlin, Humboldt-Universität zu Berlin, Brook-Taylor-Str. 2, 12489 Berlin, Germany. [3]College of Environmental & Engineering, Tongji University, 200092 Shanghai, China. [4]School of Resources and Environment, Anqing Normal University, 246011 Anqing, China. [5]School of Materials Science and Engineering, Nanyang Technological University, Singapore 639798, Singapore. [6]These authors contributed equally: Qi Dang, Wei Zhang. ✉e-mail: leizd95@tongji.edu.cn; tang1liang@shu.edu.cn

(0–2 V) and the use of an external supporting electrolyte (such as $Na_2SO_4$) to effectively suppress the recombination of photogenerated electron/hole pairs in photoanode[10,11]. These requirements inevitably increase the energy consumption of the system.

Operating photoanodes at lower bias voltages or bias-free will lead to inefficient extraction of a significant fraction of photoexcited charge carriers, which, in turn, results in recombination losses due to potential traps located along the transport pathway[12,13]. The bias voltage is critical for activating proton-electron coupling process in PEC system (Fig. 1a). Therefore, reducing the energy barrier for electron coupling and enhancing the electron coupling rate have become crucial factors. Recent research on aqueous ions batteries has demonstrated that the rapid adsorption of ions in the aqueous phase onto the cathode material can facilitate fast electron coupling[14–16]. Given the abundance of inorganic salts (such as $Na^+$, $K^+$, and $NH_4^+$) in wastewater, we speculate that a reaction pathway can be designed by modulating the type of cathode electrode material. For example, ion-electron coupling process can be substituted for proton-electron coupling process, and the oxidation-reduction potential of electrons can be adjusted. Such an approach could enable the development of ion assisted, unbiased, and high-efficiency photoelectrochemical process in PEC systems. Despite the potential advantages, the utilization of ion-assisted electron transfer pathways remains unexplored in PEC systems.

Based on the above concept, we proposed a type of IAPEC system (Fig. 1b). This system incorporates inorganic electron-ion receptor materials capable of reversible electron-ion storage as cathodes, enabling an electron-inserted ion coupling route for efficient transfer of photogenerated electrons. The spontaneous high potential difference between the IAPEC system's electron-ion receptor cathode and

photoanode not only greatly enhances the efficient separation of photogenerated electron-hole pairs. Additionally, it naturally drives the formation of hole-excited free radicals, eliminating the need for an external bias voltage. Notably, our IAPEC system demonstrated superior degradation performance in wastewater containing chloride medium. This finding suggests that, in addition to the cations accelerating the electron transfer rate, the presence of $Cl^-$ ions further enhances the efficiency and sustainability of wastewater treatment. The concept presented in this study sheds light on the potential use of abundant sodium chloride in seawater as a cost-effective additive for wastewater treatment. Overall, our research provides a strategy for designing high-performance PEC systems. The IAPEC system demonstrates significant potential for efficient and eco-friendly wastewater treatment while offering a pathway for cost-effective and sustainable resource utilization.

## Results

### Design and operation of our bias-free driven IAPEC system

To achieve a bias-free driven PEC system, the most important mission is to identify an alternative driving force provided by the applied bias to withstand the strong combination between the electron-hole pairs, thereby assisting the efficient transfer of photogenerated electrons. Here, we propose a strategy to alter the route of the photogenerated electron transfer reaction, aiming to regulate the Gibbs free energy of this progress. To achieve this, we employ PBAs as robust and reversible electron-ion acceptor, which serve as reaction sites for photoelectron-coupled inserted cations[17,18]. This enables the efficient transfer of photogenerated electrons from the proton demand in the solution to the greater abundance of sodium ions in the solution. We utilized a modified hydrothermal method to synthesize an anatase phase

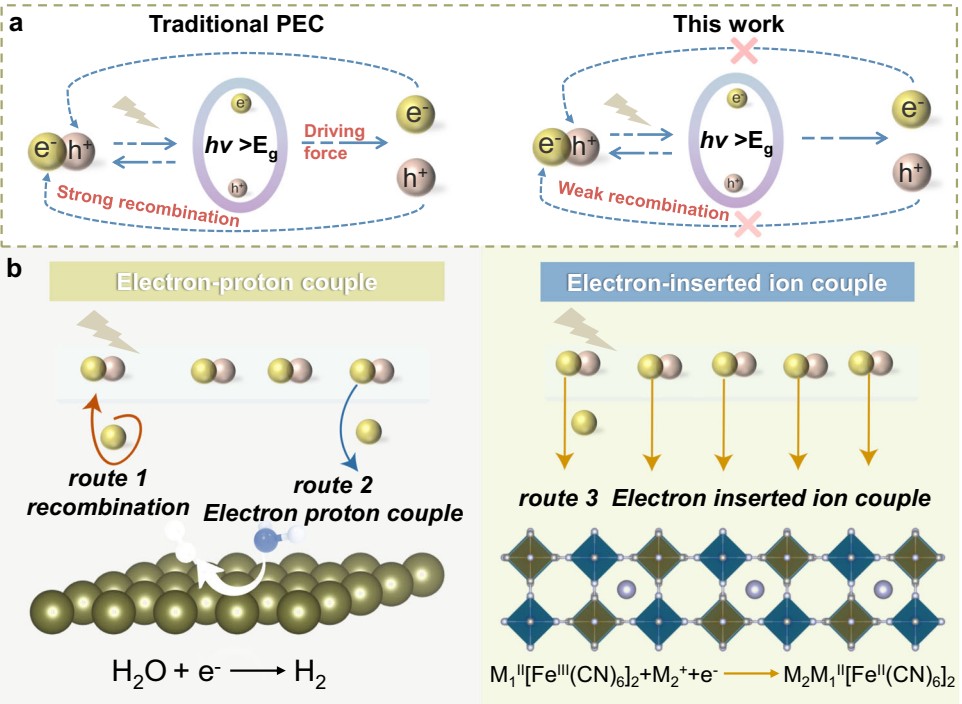

**Fig. 1 | Schematic diagram of the mechanism for of photogenerated carriers' separation pathway and the electron transfer and storage pathways behind different system. a** Traditional PEC system. **b** Our designed bias-free driven inserted ion assisted photoelectrochemical (IAPEC) system. In the conventional PEC system, the photoanode is excited by incident sunlight to generate photogenerated electrons. The photogenerated electrons are transferred to cathode through proton-coupled electron progress, the cathode often chooses an electron acceptor material (like Pt), along with hydrogen evolution reaction (HER) on the

electrode acceptor cathode. The degradation of organic pollutants is accomplished by free radicals generated by water oxidation reaction, usually hydroxyl radicals. In this work, transfer of photogenerated electrons can be carried out by an electron-coupled progress. This is enabled by the use of electron-ion acceptor material that can reversibly insert and extract of electrons/ions. Our designed IAPEC system can produce corresponding free radicals according to the change of anions in the electrolyte, and realize the synergistic degradation of organic pollutants by multiple free radicals.

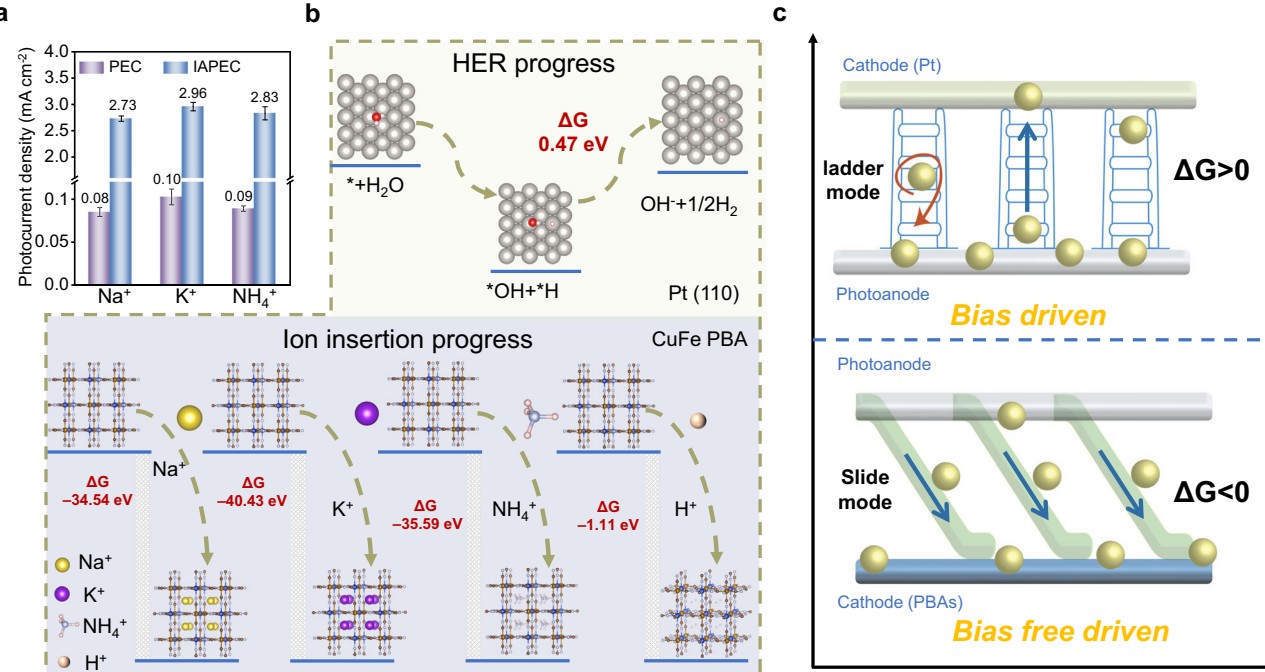

**Fig. 2 | Design and operation of bias-free driven IAPEC system. a** The transient average photocurrent density diagram of traditional PEC and IAPEC under 100 mW cm⁻² intensity with different cation aqueous solutions. **b** the theoretical calculation of the Gibbs free energy change for the intermediate species involved in the HER process, as well as the PBAs structure during the insertion process of Na⁺, K⁺ NH₄⁺ and H⁺ ions. The insets depict the corresponding structures and the proposed reaction mechanisms, along with their respective energy level diagrams. **c** Schematic diagram of the transport process of photogenerated electrons includes a ladder mode representing the bias-driven traditional PEC and a slide mode representing the bias-free drive of the IAPEC system. Error bars representing the standard deviation of three replicate measurements.

titanium dioxide (TiO₂) electrode as photoanode for our designed PEC system[19]. The detailed process for all electrode material is illustrated with the relevant descriptions in Supplementary Figs. 1–8 and the Experiment sections. The photocurrent densities generated by the IAPEC system (TiO₂-PBAs) and traditional PEC system (TiO₂-Pt) were compared in different cationic electrolytes, as depicted in Fig. 2a. The results revealed that the IAPEC system generated a photocurrent density of 2.96 mA cm⁻², which is over 30 times higher than that of the traditional PEC system (0.10 mA cm⁻²). The magnitude of the photogenerated current represents the quantity of photogenerated electrons transferred into the cathode through the external circuit. Thus, it demonstrates the IAPEC system's capability for efficient separation of photogenerated carriers under bias-free driving. We also measured the open circuit voltage (OPV) (Supplementary Fig. 9). The IAPEC system produced significantly higher voltages of 1.15, 1.19, 0.8, and 1.12 V with different types of PBAs cathode, respectively, in comparison to the 0.55 V OPV obtained by the PEC (TiO₂-Pt) system.

Considering the traditional PEC system, the photogenerated electron transfer process is primarily limited by the energy barrier for the HER process on the electrode surface[20]. To investigate this limitation, we conducted calculations on the energy barrier for a typical Pt electrode in PEC, revealing a maximum free energy of 0.47 eV for the most energy barrier step (Fig. 2b). Furthermore, we conducted a comparative analysis of the free energies associated with different ion insertions into the PBAs electrode (Supplementary Figs. 10–12). We observed a significant reduction in the free energy of the PBAs structure after ion insertion. Notably, compared to H⁺ ions, PBAs exhibited a stronger preference for the selective insertion of the three cations. As illustrated in Fig. 2b, the free energies for the insertion of Na⁺, K⁺ and NH₄⁺ ions by CuFe PBA were measured at −34.54, −40.43, and −35.59 eV, respectively, while the free energy associated with inserting H⁺ ions was notably higher at −1.11 eV. This observation suggests that in traditional PEC systems, where the

Gibbs free energy exceeds 0, the process of photogenerated electrons flowing into the electrode requires an external bias like a ladder mode to drive the process (Fig. 2c). This process is prone to electron-hole recombination and slow electron transfer, limiting the photocurrent density. In contrast, our IAPEC system offers a solution to the inefficiencies observed in traditional PEC systems. By operating in a slide mode with a Gibbs free energy much lower than 0, our system facilitates the smooth flow of photogenerated electrons into the electrode without the need for a biasing voltage (Fig. 2c). This slide mode operation is a promising approach to achieve efficient and carrier-separating process, thus effectively improving the performance of PEC systems.

To better understand the changes of PBAs electrode for auxiliary transfer process of ion-electron coupling, the interaction mechanism was explored in detail. The typical crystal structure of PBA is shown in Fig. 3a. X-ray photoelectron spectroscopy (XPS) analysis revealed the evolution of Na 1 s peak appeared at 1072.4 eV, and the area of Fe (III) 2p region gradually decreases while the area of Fe (II) 2p region increases, indicating that most Fe (III) are reduced to Fe (II) after the Na⁺ insertion process of CuFe PBA. (Supplementary Figs. 13 and 14)[21]. The ex situ X-ray absorption near-edge structure (XANES) of Fe K-edge spectra are shown in Fig. 3b. By taking a closer look at the dipole forbidden, the Fe pre-edge peaks of shift significantly towards lower energy, demonstrating that most of the Fe is nearly in the +2 state (Supplementary Table 1)[22]. Combined valence analysis shows that the average valence states of Fe are 2.9 and 2.1. (Fig. 3c and Supplementary Table 1). The Extended X-ray adsorption fine structure (EXAFS) analysis revealed that Fe-C bond did not shift significantly after Na⁺ insertion, and Fe-C-N bond length was shortened, which may be due to lattice shrinkage caused by bond distortion (Fig. 3d–f and Supplementary Table 2)[23]. This result was also confirmed in XRD, when after the light-driven Na⁺ ions is inserted into CuFe PBA (Supplementary Fig. 15)[24]. The crystal structure of CuFe PBA always maintains a face-centered cubic

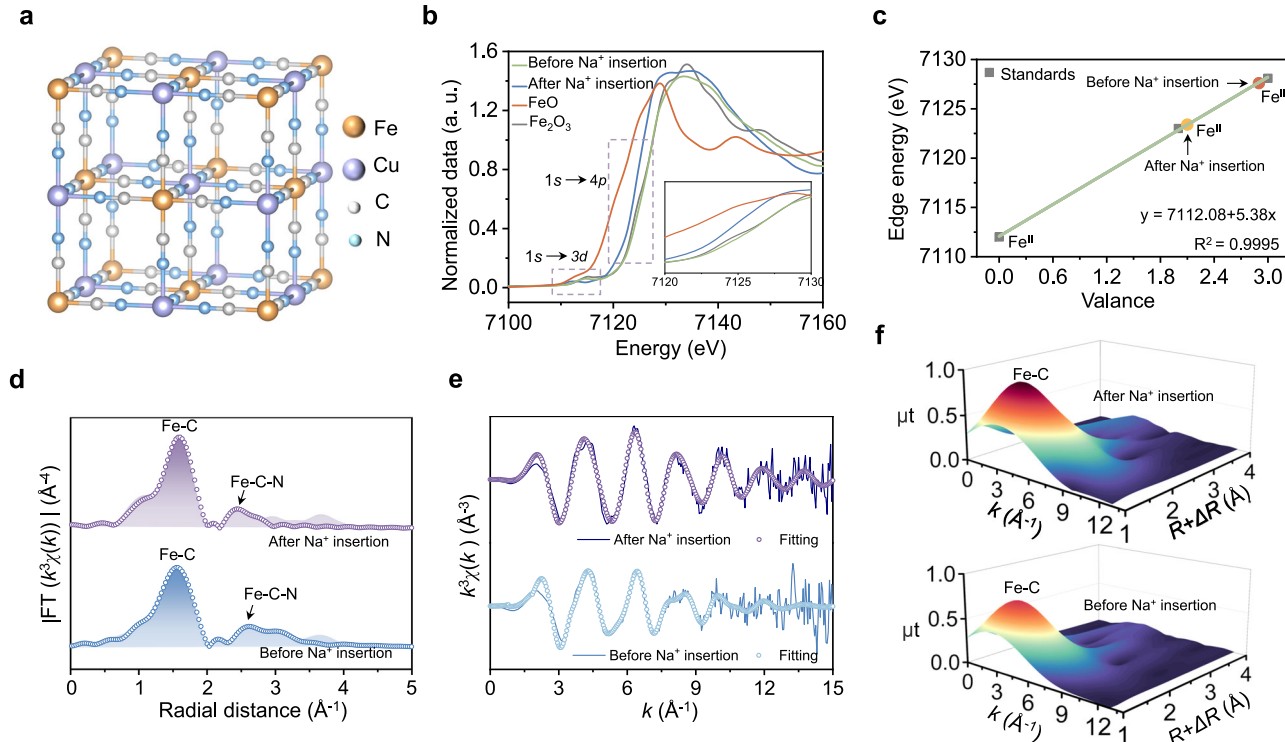

**Fig. 3 | Mechanism study of PBA electrode in ion-electron coupled assisted transfer process. a** Schematic diagram of typical crystal structure of PBA. **b** Fe *K*-edge XANES spectra. **c** Average oxidation state fitting curve. **d** EXAFS analysis of the *k*-space spectrum **e** EXAFS analysis of the *r*-space spectrum and corresponding fitting curve. **f** Wavelet transform contour plots of Fe *K*-edge at *r* space.

structure with the insertion of Na+ ions, and the position of the diffraction peak shifts to a higher angle of the diffraction peak, indicating the slightly shrink of framework. To further investigate the changes in the crystal structure of CuFe PBA before and after the insertion of Na+ ions, we employed synchrotron powder X-ray diffraction to acquire high-resolution structural data, allowing for a refined analysis of the crystallographic information. Supplementary Fig. 16 depicts the refined crystal structure before and after the insertion of Na+ ions. CuFe PBA retains its Fm-3m structure after insertion of Na+ ions, leading to a change in the lattice parameter from 10.110 to 10.076 Å. This can be attributed to superior stability of CuFe PBA, which has an ultra-low strain open frame structure[25]. Raman spectra characterization can be seen that after Na+ ions insertion, the characteristic peak of the cyanide shifted in lower wavenumber peaks. Due to the frequency of the cyanide stretching vibration mode is sensitive to the surrounding chemical environment, so the cyanide coordinated with Fe (II) shows a relatively lower wavenumber peaks than the cyanide coordinated with Fe (III) (Supplementary Fig. 17)[26].

**Investigation of IAPEC system degradation performance for saline wastewater**

First, we conducted a comparison of various systems in degrading methylene blue (MB) under identical experimental conditions, including photocatalysis (PC), electrocatalysis (EC), traditional PEC (A bias voltage of 0 V or 1 V applied), and IAPEC in sulfate and chloride medium, respectively (Fig. 4a). The results showed that IAPEC exhibited the best degradation performance in chloride medium, among the different systems[27]. And the corresponding pseudo-first-order rate constant of IAPEC in chloride medium reached as high as 0.5 min−1 in 60 min, compared to only 0.24, 0.012, 0.005, 0.004, and 0.003 min−1 for IAPEC (sulfate), PEC (1 V), PEC (0 V), EC and PC respectively (Supplementary Fig. 18). Therefore, we further investigated the degradation performance of MB under different NaCl concentrations medium. When the concentration of NaCl increased from 0.01 M to 2 M, the

overall degradation efficiency of MB increased (Fig. 4b). Notably, when the concentration was 0.1 M or higher, all systems achieved 100% degradation within 20 min. Impressively, even when the concentration is as low as 0.01 M, the degradation can still reach ~80% within 60 min. This suggested that IAPEC could effectively degrade organic pollution under different ionic strength conditions especially when the solution with high salinity. Transient photocurrent measurements were conducted through intermittent simulation of sunlight irradiation to assess the effects of ionic strength on charge separation and transfer rates (Supplementary Fig. 19). When the concentration of sodium chloride increased from 0.01 M to 0.5 M, the photocurrent density of the conventional PEC system increased from 0.04 to 0.05 mA cm−2, while that of the IAPEC system increased from 1 to ~1.6 mA cm−2. This phenomenon could be attributed to the fact that higher ionic strength facilitates faster conductivity in the PEC system. In contrast, the IAPEC system exploits the additional function of cations in promoting photogenerated electron transfer, which is complemented by the cathode PBAs material that provides ample intercalation sites to accelerate the electron-ion transfer process.

We also evaluated the degradation performance of IAPEC system based on four different PBAs cathodes for MB (Fig. 4b). No external bias was adopted. The exciting thing is that all the systems achieved 100% degradation within 20 min. This indicates that the mechanism of ion-electron coupling in our designed PEC system can be widely applied to more ionic electrode materials. Thereafter, to broaden the light harvesting range of the IAPEC system, we employed a bismuth vanadate (BiVO4) photoanode with a visible light absorption band gap[28], in addition to the previously selected TiO2 photocatalyst which absorbs UV light. The characterization of the BiVO4 electrode is presented in Supplementary Fig. 6. It was observed that BiVO4-PBAs system had excellent degradation effect on MB compared with TiO2-PBAs system (Fig. 4c). This above result indicated that our IAPEC system can achieve efficient degradation of pollutants under visible light irradiation.

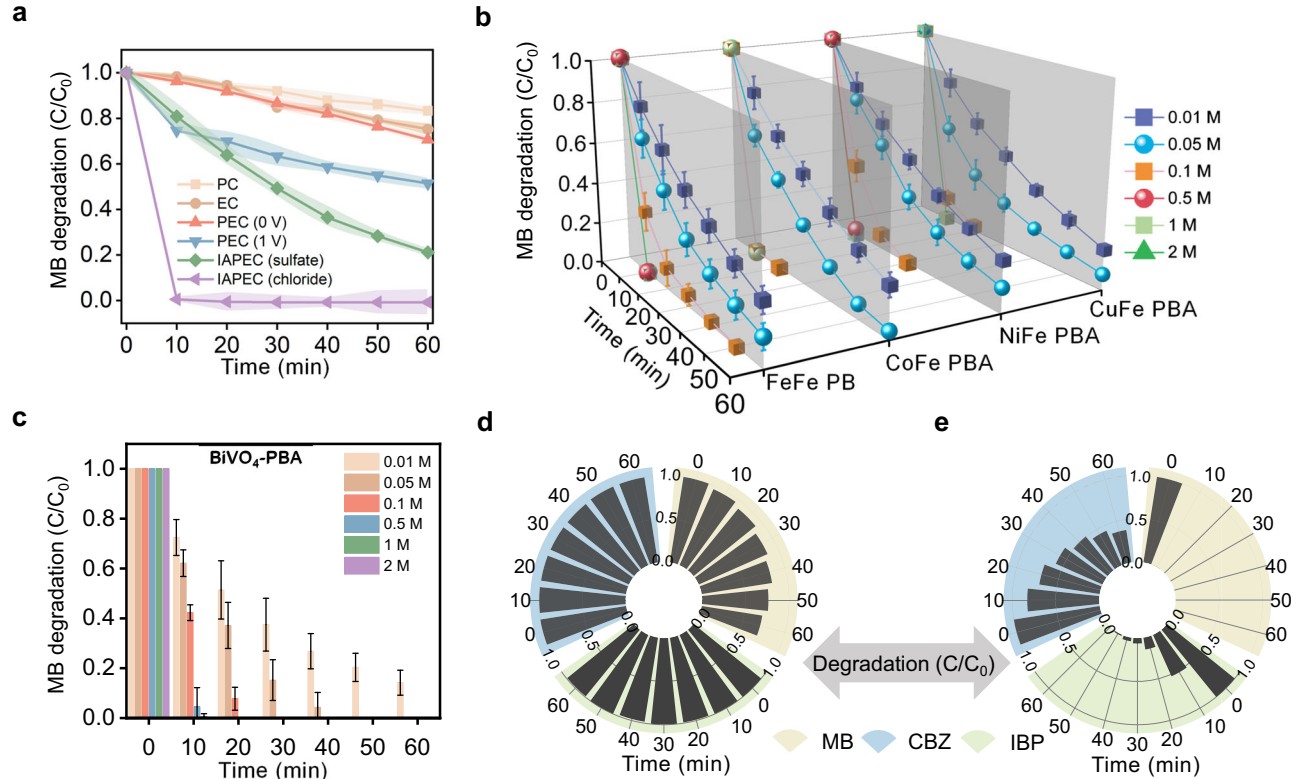

**Fig. 4 | Degradation performance of IAPEC system in simulated saline wastewater.** **a** The degradation performance for MB by different system. **b** The degradation 3D line diagram of MB pollutants by IAPEC system under different PBA counter electrode. **c** the degradation of MB at different concentrations under BiVO$_4$-PBA photoanode system. **d** The radial histogram of degradation of different pollutants over time by PEC system. **e** The radial histogram of degradation of different pollutants over time by IAPEC system. (Experimental conditions: 60 mL simulated saline sewage, initial pH = 6.0 ± 0.1, simulated solar light illumination 100 mW cm$^{-2}$, [NaCl] = 0.1 M, [MB]$_0$ = 10 ppm. MB, IBP, CBZ). Error bars or error bands representing the standard deviation of three replicate measurements.

In addition to the traditional organic dye pollutant MB, two representative emerging pollutants, ibuprofen (IBP) and carbamazepine (CBZ) were also selected as model pollutants to assess the generality of the IAPEC system in the removal of different types of pollutants. In the traditional PEC system, MB was degraded by 30% after 1 h, while CBZ and IBP were barely degraded (Fig. 4d). In contrast, the IAPEC system showed significant advantages in degrading the three pollutants, achieving a degradation rate of 90% within 30 min (Fig. 4e). Notably, as the MB concentration increased from 5 ppm to 30 ppm, the IAPEC system achieved remarkable performance by effectively removing over 99% of MB within 10 min (Supplementary Fig. 20). The IAPEC system exhibited excellent light receiving ability even in in the presence of high concentration dye environment, and the IAPEC system degradation performance was not affected by the chromaticity of the dye. However, the degradation rate of 10 ppm MB by traditional PEC in 60 min was only 30%. The IAPEC system achieved 99% degradation within 20 min as the IBP concentration increased from 1 ppm to 5 ppm (Supplementary Fig. 21). Even at a higher concentration of 10 ppm, the system still achieved over 90% degradation within 50 min. In contrast, the conventional PEC system exhibited only 1.8% degradation rate for 10 ppm IBP after 60 min. Similarly, the IAPEC system achieved complete degradation of 1 ppm CBZ within 10 min, with degradation rates of 5 and 10 ppm CBZ reaching 81% and 63% within 1 h, respectively (Supplementary Fig. 22). In contrast, the PEC system displayed negligible degradation rates for CBZ. We further targeted a broad range of recalcitrant chemicals with our pollutant model, including Bisphenol A (BPA), 4-Chlorophenol (4-CP), Perfluorooctanoic Acid (PFOA), Sulfamethoxazole (SMX), and Cellulose Acetate Propionate (CAP), as shown in Supplementary Fig. 23. After a 60 min treatment in the PEC system, the degradation rate of other pollutants was basically ignored except for 50% degradation of CAP. In contrast, in the IAPEC system, except for PFOA, all other pollutants reached 80% degradation after 1 h treatment.

We assessed the total organic content (TOC) of pollutants treated by the IAPEC system (Supplementary Fig. 24). The findings revealed that the IAPEC system achieves an impressive mineralization efficiency, reaching up to ~50%. Particularly significant is the observation that pollutants with higher organic carbon content, such as PFOA, also achieved a commendable mineralization efficiency of about 40%. These results emphasize that the IAPEC system excels not only in efficiently remediating pollutants but also in its potential to mitigate secondary pollution following pollutant degradation. These results incontrovertibly demonstrate that the IAPEC system possessed significant advantages in degrading emerging organic pollutants than the traditional PEC system, especially in refractory organic pollutants.

Additionally, we monitored the concentration of the toxic byproduct chlorate during the degradation of MB in the IAPEC system using different photoanode. As illustrated in Supplementary Fig. 25, the results indicated that when employing a TiO$_2$ photoanode, the concentration of ClO$_3^-$ was ~0.5 mg L$^{-1}$. In contrast, for the BiVO$_4$ system, the ClO$_3^-$ concentration was around 0.15 mg L$^{-1}$ (with a health limit of 0.7 mg L$^{-1}$), and in both systems, the ClO$_4^-$ concentration was undetectable[29]. In contrast, IAPEC system demonstrates exceptional pollutant degradation while exhibiting a superior inhibitory effect on the generation of toxic byproducts (Supplementary Table 3).

Reusability performance is also as a vital indicator to evaluate the stability of the PEC system. As depicted in Supplementary Fig. 26, the degradation rate of MB only slightly decreased after 20 cycles but still remained high at 92.8%. The degradation rates of IBP and CBZ were found to be 50% and 20% within 10 min (Supplementary Figs. 27 and 28).

After 20 cycles, the degradation efficiencies of different pollutants decreased by less than 5% compared to the initial cycle, indicating that the IAPEC system has good cyclic stability for pollutant degradation. The XRD patterns of the PBAs electrode used for cycle experiment were found to be similar to those of the initial sample (Supplementary Fig. 29).

Recently, the anti-interferent factor of PEC system was defined as an important index to assess the practicability of the system[30]. Here we select four typical environmental parameters to evaluate the performance of the IAPEC system in the actual environment. The pH value of wastewater is an important system parameter in PEC because it affects the surface charge distribution of semiconductors, the morphology of organic compounds, and the degradation pathway. The pseudo-first-order rate constant of MB degradation decreases from 0.48 to 0.21 min$^{-1}$ as pH value increases from 2 to 10 (Supplementary Fig. 30). In general, the dissociation and distribution of free chlorine substances in solution are related to pH value. When pH value is 2, the active chlorine species in solution are mainly $Cl_2$ and HClO, while 3<pH<8 and pH > 8.0, HClO and ClO$^-$ dominate, respectively[31,32]. On the one hand, the increase of $Cl_2$ (aq.) content can promote the formation of chlorine-free radical (such as Cl•, $Cl_2^-$•, ClO•); on the other hand, $Cl_2$ (aq) ($E_0 = 1.36$ V vs. SHE) and HClO ($E_0 = 1.49$ V vs. SHE) are more oxidizing because they have a higher standard redox potential than ClO$^-$ ($E_0 = 0.89$ V vs. SHE)[33,34]. By monitoring the pH values under different initial conditions (Supplementary Table 4), it can be found that the pH value of the system is maintained at about 2. The decrease in pH is mainly due to the hole oxidation of $H_2O$ and chloride ions resulting in the production of hydrogen ions and free chlorine species (FCS) (Supplementary Fig. 31)[35]. This suggests that our system has the ability to autonomously regulate the pH to create conditions favoring the presence of highly reactive free radicals, thereby enhancing the efficiency of pollutant degradation.

Apart from pH, as the temperature of the simulated wastewater increases from 15 °C to 75 °C, the degradation rate of MB exhibits a trend of initially increasing and then decreasing (Supplementary Fig. 32). The increase in temperature promotes the process of ion intercalation into the PBAs lattice, thereby improving the transfer efficiency of photogenerated electrons. However, as the temperature continues to escalate, the solubility and stability of key active species, such as $Cl_2$ and HClO, exhibit a decline[36,37]. This diminished availability of these species hampers the population of effective free radicals, leading to a decline in the efficiency of pollutant degradation. In summary, the observed temperature-dependent degradation kinetics of MB can be attributed to the interplay between these two competing factors: the initial amplification of the ion insertion process and the subsequent depletion of effective free radicals caused by the decreased solubility and stability of active species. This ultimately results in a reduction in the efficiency of pollutant degradation. The insights gained from our study provide valuable information for the development of effective strategies for pollutant removal in various environmental applications.

Considering that the actual wastewater is a multi-ion coexistence environment, we discussed the effect of the presence of typical cations on the degradation rate of the IAPEC system. Keep a certain concentration of Cl$^-$ ions, when the electrolytes were NaCl, KCl, $CaCl_2$, and $MgCl_2$, the pseudo-first-order rate constants of MB degradation under the four electrolytes were NaCl (0.5 min$^{-1}$) > KCl (0.4 min$^{-1}$) > $CaCl_2$ (0.1 min$^{-1}$) > $MgCl_2$ (0.2 min$^{-1}$; Supplementary Fig. 33). The magnitude of the photogeneration current further substantiates this perspective (Supplementary Fig. 34). One aspect pertains to the lower concentrations of $Ca^{2+}$ and $Mg^{2+}$ ions observed at the equivalent Cl$^-$ ions concentration. Additionally, the reduced concentration of additional charge (represented by the ratio of normal charge to ionic radius) for $Ca^{2+}$ and $Mg^{2+}$ ions pose challenges for timely electron-ion coupling, consequently resulting in a slower rate of photoelectron transfer.

In addition to studying the impact of various cations, we also conducted extensive investigations into the effects of diverse anions found in wastewater on the efficiency of pollutant degradation. As shown in Supplementary Fig. 35, after 60 min of degradation at the same concentration, the NaCl medium system exhibited the fastest degradation rate, with a pseudo-first-order rate constant of 0.5 min$^{-1}$, which was about 20 times higher than that of $Na_2SO_4$ (0.025 min$^{-1}$) and $NaNO_3$ (0.024 min$^{-1}$), respectively. This phenomenon could be attributed to the interaction between Cl$^-$ ions and the generated holes, leading to the generation of highly reactive chlorine species. This finding suggests that the IAPEC system has the potential to generate different types of free radicals based on the anions present in wastewater. By regulating the composition of anions in wastewater, it is anticipated that the degradation efficiency of IAPEC can be further enhanced.

The presence of natural organic matter (NOM) may affect the formation and transformation of free radicals, thus affecting the effect of the system. NOM, a ubiquitous component found in natural water, can affect the formation and transformation of free radicals, thus affecting the effect of the system[38]. We focused on investigating the influence of humic acid (HA), a common constituent of NOM, on MB degradation using the IAPEC system. Increasing HA concentration from 5 ppm to 50 ppm, the pseudo-first-order rate constant of MB removal decreases from 0.5 min$^{-1}$ to 0.1 min$^{-1}$ (Supplementary Fig. 36). This phenomenon can be attributed to two aspects. Firstly, HA absorbed UV light, resulting in the reduction of UV light absorbed by $TiO_2$ photoanode, thus generating fewer chlorine free radicals[39,40]. Secondly, the functional groups such as carboxyl, hydroxyl, and carbonyl on HA molecules can react with active chlorine substances, forming chlorine-containing compounds that competed with the degradation of MB[41,42].

To assess the degradation potential of the IAPEC system in real saline aqueous medium, we selected secondary effluents collected from a coal manufacturing plant in Inner Mongolia, China (43.46°, 113.12°) were spiked with MB and CBZ as the target pollutant without externally adding any electrolyte and without any further treatment like adjusted the pH (Fig. 5a). Upon exposure to simulated sunlight, both MB and CBZ exhibited removal rates exceeding 90% within 30 min (Fig. 5b). To validate the system performance, we assembled a prototype device and subjected it to actual solar for 5 h, resulting in degradation rates of ~50% for MB respectively. More exciting, the IAPEC system also demonstrated high efficiency in simultaneously removing other pollution indicators from the actual sewage. The treated wastewater was analyzed for COD (chemical oxygen demand) and TDS (total dissolved solids) using a Hash water quality analyzer and a conductivity test. The results indicate that the system was able to reduce the COD and TDS of the wastewater by 5800 and 120 mg L$^{-1}$, respectively, after 1 h of treatment, and by 13000 and 860 mg L$^{-1}$, respectively, after 5 h of treatment (Fig. 5c). This is attributed to the ion-assisted photoelectron coupling transfer process employed by the IAPEC system, where the degradation of pollutants is accompanied by the transfer of ions in wastewater. The findings underscore the effectiveness of this method in treating complex industrial wastewater containing refractory organic pollutants.

Drawing upon our above results, the IAPEC system exhibited notable efficacy in degrading pollutants within a NaCl medium. As such, our proposed concept underscores the potential utility of utilizing the abundant NaCl found in seawater as a cost-effective additive for wastewater treatment. In light of this, we selected authentic seawater samples from three different regions of China (Fig. 5d) and spiked them with MB, IBP, CBZ, BPA, 4-CP, SMX, PFOA, and CAP at a concentration of 10 ppm as target pollutants. No additional electrolyte was introduced, and neither pH adjustments nor further treatment procedures were undertaken. Based on the data presented in Fig. 5e, the degradation of the eight pollutants was observed to be completed

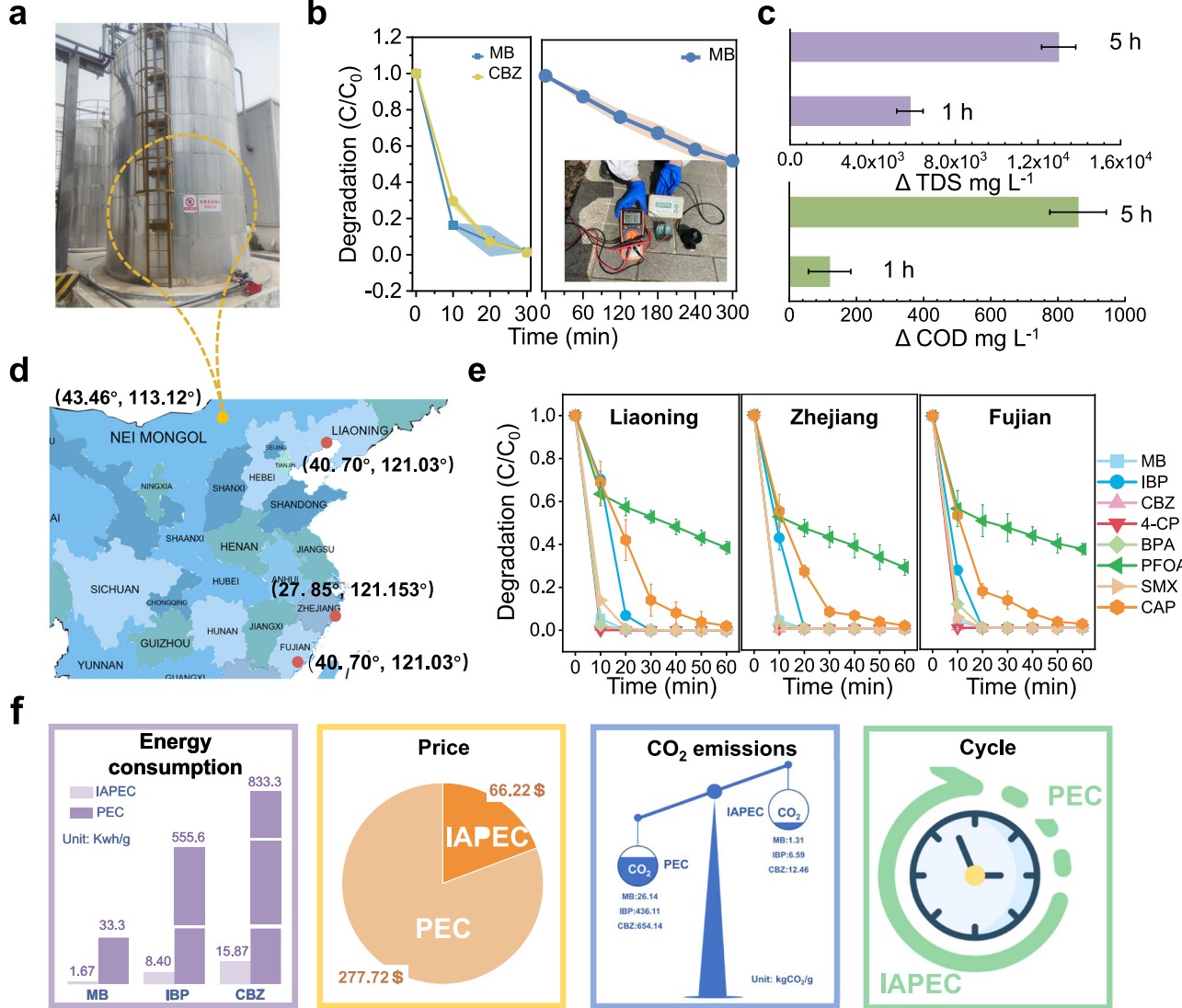

**Fig. 5 | Degradation performance of IAPEC system in real saline wastewater. a** A digital photo of a real wastewater pond at a coal chemical plant in Inner Mongolia (China, 43.46°, 113.12°). **b** Diagrams of degradation efficiency of MB and CBZ by IAPEC system under simulated (left) and MB under real solar (right) in real saline wastewater background. The inset picture is a homemade prototype device used for real solar degradation system. **c** The removal effect of TDS and COD in real wastewater by IAPEC system. **d** The map location of different water sample collection points. **e** The degradation 3D line diagram of three pollutants by the IAPEC system in three different seawater backgrounds. **f** Sustainability evaluation of the system from four aspects: energy consumption, price, carbon emissions and operating cycle. Error bars or error bands representing the standard deviation of three replicate measurements.

within 30 min across all three types of seawater backgrounds, except for CAP and PFOA. Remarkably, CAP degradation was nearly complete within 1 h, while the degradation rate for the highly stable PFOA within the same timeframe ranged approximately around 65%. To the best of our knowledge, no previous instances exist where the utilization of real seawater enhances the degradation of organic pollutants in PEC systems.

The sustainability of wastewater treatment technology systems is of great significance for the realization of carbon neutralization goal. As shown in Fig. 5f, we compare the IAPEC with the traditional PEC from the four aspects of energy consumption, price, carbon emission, and operation cycle of the technology, and the comparison results of the four aspects show obvious technological increment.

These findings contribute valuable insights into the utilization of locally available resources for wastewater treatment and present an opportunity for the development of environmentally friendly and sustainable wastewater treatment processes with reduced carbon footprint.

**Mechanism and degradation pathway insight into IAPEC system**

In order to identify the generation of reactive species in IAPEC system, the reactive species involved in MB degradation and their contribution to MB degradation can be obtained by adding different free radical scavengers to the NaCl solution (Supplementary Fig. 37)[43]. There was almost no change in the degradation rate after quenching •OH with NB (nitrobenzene), indicating that the main reactive species involved in degradation was not •OH. After adding TBA, the degradation rate of MB was significantly slow, indicating that the degradation process was mainly involved by chlorine free radicals. The addition of $Na_2S_2O_3$ quenched all active substances, indicating that direct oxidation from holes could not completely remove pollutants, further indicating the important role of reactive chlorine species in IAPEC chlorine medium system. The addition of EDTA-2Na has been observed to maintain the degradation rate of MB at a consistent level with all the reactive species. This observation indicates that the reactive species involved in the degradation process are derived either directly or indirectly from the oxidation of holes.

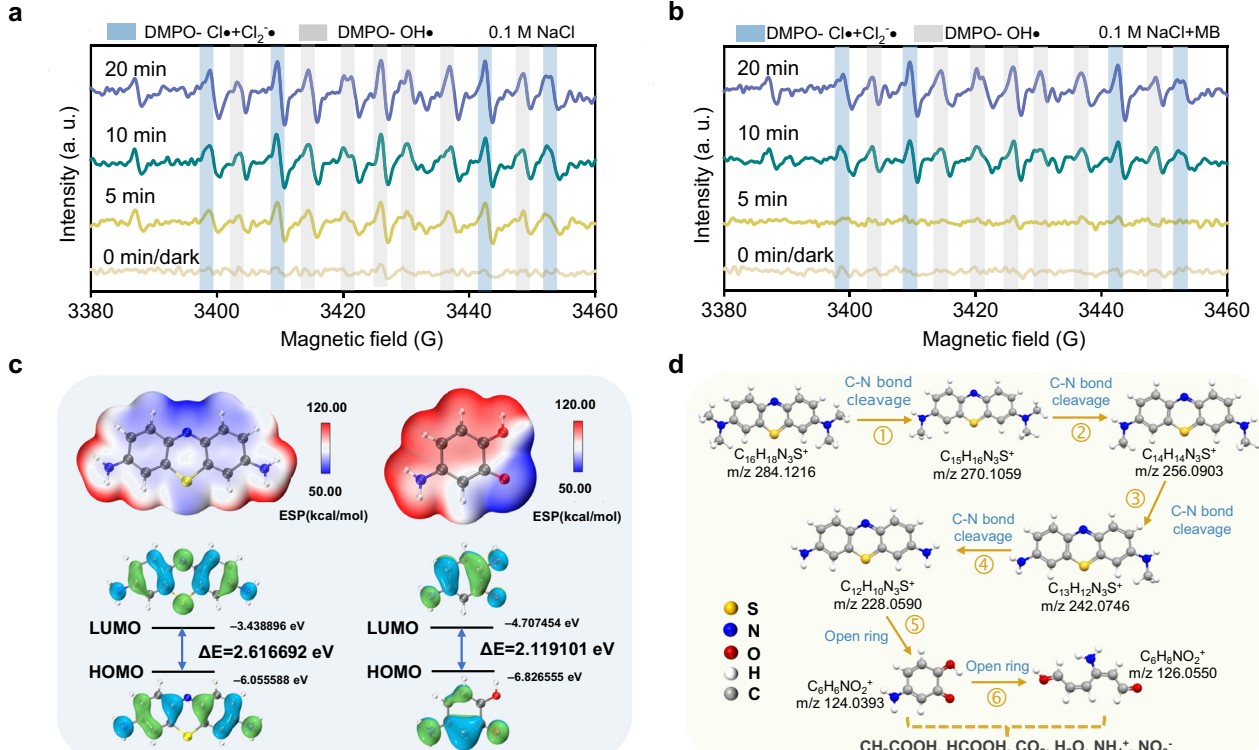

**Fig. 6 | Degradation mechanism and paths studies for the IAPEC system.**
**a** ex situ ESR tests of IAPEC system under 0.1 M NaCl as electrolyte. **b** ex situ ESR tests of IAPEC system under 0.1 M NaCl and 10 ppm MB as electrolyte. **c** DFT calculation. (The red and blue indicate the regions with the most negative potential (electron-rich) and positive potential (electron-poor), respectively. The negative and positive regions are promising sites for reduction and oxidation reactions, respectively. The energy difference between HOMO and LUMO is the band gap. The smaller the band gap is, the more easily the molecule is excited. The $f^0$, $f^+$, $f^-$ indicate the response of different atoms to the disturbance caused by the chemical attack. **d** Degradation path analysis.

The type of active species in the IAPEC system was more intuitively determined by exsitu electron spin resonance (ESR) spectroscopy. As depicted in Fig. 6a, when the IAPEC system exposure to solar in NaCl medium triggers the spontaneous generation of chlorine free radicals, as evidenced by the detection of 11 peaks of typical chlorine free radical and hydroxyl radical signals after 5 min illumination. Conversely, no such free radicals were observed in dark conditions or in the conventional PEC system (Supplementary Fig. 38). With the addition of pollutants, the signals of chlorine radicals and hydroxyl radicals were slightly weakened and delayed, which originated from the generated free radicals are indeed involved in the degradation of MB (Fig. 6b). Several studies have investigated the kinetics and mechanisms of reactions involving chloride radicals, including Cl• and $Cl_2^-$•, with various organic compounds. In the context of photolysis, the formation of Cl• through the reaction of free chlorine has been found to possess a redox potential of 2.4 V vs. SHE[44], which can further facilitate the degradation of contaminants through various mechanisms, including dehydrogenation, single electron transfer, or addition of unsaturated bonds. Although $Cl_2^-$• ($E_0$ = 2.13 V vs. SHE)[45] and ClO• ($E_0$ = 1.39 V vs. SHE)[46,47] are secondary free radical species, they also play an important role in the removal of emerging organic pollutants. Meanwhile, we comprehensively analyzed the pathways for generating active species within the IAPEC system in sulfate media, utilizing insights from ESR, quenching experiments, and molecular probe experiments (Supplementary Figs. 39–43)[48–50]. As presented in Supplementary Fig. 43, the degradation rate of oxalic acid (0.03 min$^{-1}$) was notably approximately twice that of nitrobenzene (0.0164 min$^{-1}$) and phenol (0.0179 min$^{-1}$). Our chemical kinetic model calculations validated the prominent role of h$^+$ (45%), while the quenching experiment results underscored the contributions of •OH (24.6%) and •$SO_4^-$ (30.1%). Furthermore, our calculations based on the chemical kinetic model emphasized the predominant role of chlorine free radicals (Cl•, $Cl_2$•, ClO•) (71.46%), followed by •OH (18.89%) and free chlorine substances (ClO$^-$/HClO (8.4%)) (Supplementary Fig. 44). This finding underscores the remarkable potential of the IAPEC system as a promising approach for environmental remediation, as it can spontaneously generate chlorine free radicals in the presence of NaCl medium, which may explain its remarkable performance in pollutant degradation.

A density functional theory (DFT) theoretical calculation was conducted to evaluate the Fukui function and predict the reactive sites on the molecule. Subsequently, a comprehensive analysis of the structure-property relationship of the organic molecule was performed[51,52]. The molecular structures of $C_{12}H_{10}N_3S^+$, $C_6H_6NO_2^+$, highest occupied molecular orbital (HOMO), lowest unoccupied molecular orbital (LUMO) and ESP (electrostatic potential) are shown in Fig. 6c, detailed data of Fukui electrophilic index ($f^-$), nucleophilic index ($f^+$), high radical index ($f^0$) and condensed dual descriptor (CDD) are shown in Supplementary Table 5, 6. The distribution of ESP and charge can provide useful information for reaction center analysis. As shown in Supplementary Figs. 45–48, N19 (0.0622), S20 (0.0719) on $C_{12}H_{10}N_3S^+$ and C2 (0.1232) and O12 (0.1434) on $C_6H_6NO_2^+$ have higher $f^0$ values than other sites. This indicates that the benzene rings of $C_{12}H_{10}N_3S^+$ and $C_6H_6NO_2^+$ molecules are vulnerable to attack. Meanwhile, the degradation pathway of MB was analyzed byLiquid Chromatograph Mass Spectrometer (LC-MS) product identification results and DFT results (Supplementary Fig. 49). By analyzing the structure and type of the products obtained, it is speculated that the degradation pathway mainly comes from the following reaction channels (Fig. 6d): The degradation behavior of the initial substrate (Protonated excimer ion peak [M]$^+$ in ESI-MS is m/z 284) MB is mainly manifested in the process of C-N bond fracture and methyl removal. The C-N bond of

$C_{16}H_{18}N_3S^+$ (m/z 284) was attacked by chlorine free radicals, and the C-N bond was gradually broken and demethylated to obtain the products $C_{15}H_{16}N_3S^+$ (m/z 270), $C_{14}H_{14}N_3S^+$ (m/z 256). $C_{13}H_{12}N_3S^+$ (m/z 242) and $C_{12}H_{10}N_3S^+$ (m/z 228). In $C_{12}H_{10}N_3S+$ (m/z 228), N19 (0.0622) and S20 (0.0719) were attacked by chlorine-free radicals and then oxidized and cleaved to produce $C_6H_6NO_2^+$ (m/z 124). The groups of C2 (0.1232), C4 (0.0888), and O12 (0.1434) are further oxidized to $C_6H_8NO_2^+$ (m/z 126) under the action of chlorine free radicals. The above products are further degraded to produce small molecules of acetic acid, formic acid, $H_2O$, $CO_2$, $NO_3^-$, $NH_4^+$, etc., thus achieving complete degradation of the compound.

In summary, we have manipulated the photoelectron-accepting pathway in the counter electrode by incorporating PBAs as the electrode material. This approach has effectively reduced the energy barrier for photogenerated electron transfer and enhanced the separation of photogenerated carriers. By transforming the conventional ladder mode of PEC into the slide mode of the IAPEC system, we paved the way for a bias-free ionassisted strategy that contribute to efficient and sustainable wastewater treatment. These results demonstrated significant improvements in degradation efficiency, cost-effectiveness, carbon emissions, operation cycle, and resistance to environmental interference when compared to the traditional PEC system, particularly in the degradation of 8 representative pollutants (MB, IBP, CBZ, BPA,4-CP, PFOA, SMX, and CAP). Moreover, the coupling of photogenerated electrons and ions in this system provided a convenient pathway for the generation of highly reactive free radicals, such as chlorine radicals, opening up the possibility of using seawater as an inexpensive additive in wastewater treatment. This work lays the foundation for further exploration and optimization of IAPEC systems, with immense potential for addressing the global challenge of wastewater treatment in a sustainable manner.

## Methods

General information. Except noted, all chemicals used were of analytical grade. Deionized water was used in all experiments.

### Characterization

The morphologies of PBAs, $TiO_2$, and $BiVO_4$ were studied by a scanning electron microscope (SEM) (Zeiss Gemini SEM 300). The crystalline structure of PBAs, $TiO_2$ and $BiVO_4$ was determined by the XRD. XRD patterns were recorded with a Bruker D8 Advance X-ray diffractometer with Cu-$K\alpha$ as the diffraction target ($\lambda = 0.154178$ nm). XPS (Thermo Scientific K-Alpha) was used to obtain and analyze the binding energy of the PBAs samples. UV-vis diffuse reflectance spectra (UV-vis DRS) was measured by an UV-vis spectrometer (UV-2600, Shimadzu, Japan). IBP, CBZ, BPA, 4-CP, SMX and CAP in the solution were analyzed by high performance liquid chromatography (Ultimate 3000, Dionex). In the experiment, the concentration of IBP was determined, with 25% $H_3PO_4$ (0.1%) and 75% acetonitrile as the mobile phase gradient and the UV detection wavelength was 230 nm. The concentration of CBZ was determined with 20% acetonitrile and 80% deionized water as the mobile phase gradient and the UV detection wavelengths were 237 nm. Determination of BPA concentration, with 65% acetonitrile and 35% deionized water as the mobile phase gradient and the UV detection wavelength was 227 nm. The concentration of 4-CP was determined with 50% acetonitrile and 50% deionized water were used as the mobile phase gradient and the UV detection wavelength was 279 nm. The concentration of SMX was determined with 70% formic acid and 30% methanol as mobile phase gradients and the UV detection wavelength was 265 nm. CAP concentration was determined with 70% formic acid and 30% methanol as mobile phase gradients and the UV detection wavelength was 265 nm. A high-performance liquid chromatography (HPLC, Waters I class- AB 5500) coupled with a mass spectrometer (MS) was utilized to determine the concentration of PFOA permeated from solution. During the experiment, using a mobile phase gradient

comprising 10% methanol and 90% water solution with 0.1 wt% formic acid. The methanol permeation rate was gradually increased from 10% to 95% within 1 min, held for 4 min, then decreased to 10%, followed by 2 min of stabilization. In a negative mode, the Quadrupole -time of flight MS was operated using an electrospray ionization (ESI) interface. Chlorate ($ClO_3^-$) content was analyzed using an ion chromatograph (Thermo Scientific ICS-5000 +). The total organic carbon content of the eight model pollutants before and after degradation was monitored using a TOC analyzer (TOC multi-N / C 3100). The concentration of FCS was measured by N, N-diethyl-1, 4-phenylenediamine (DPD) coloration method using UV-Vis spectrophotometer (UV 7504/PC), which has been widely reported in previous studies. The species and signal intensity of free radicals were determined by EPR. (Bruker EMXnano). Quenching experiments were carried out with tert–butanol (TBA, •OH, Cl•, $Cl_2^{-}•$, ClO•), $NaHCO_3$ (•OH, Cl•, $Cl_2^{-}•$), nitrobenzene (NB, •OH), $Na_2S_2O_3$(•OH, Cl•, $Cl_2^{-}•$, ClO•, HClO) and EDTA-2Na ($h^+$). The intermediates and degradation paths of MB were analyzed by LC-MS (Agilent 1290 UPLC, Q-TOF 6550, waters BEH C18 2.1*100 mm 1.7um). In the molecular probe experiment, oxalic acid (OA), nitrobenzene (NB) and phenol were used as probes for direct oxidation, •OH and •OH / $SO_4^{-}•$, respectively. The exposure and steady-state concentration of free radicals can be evaluated according to the equation.

The powder XRD method on the powder diffraction beamline at the Australian Synchrotron at a wavelength ($\lambda$) of 0.6888 Å respectively, calibrated with the standard reference material (LaB6 660b). The obtained powder data from the synchrotron were indexed and refined with TOPAS 5 (Bruker) software. All samples were also used to determine their structure by Raman spectroscopy (Renishaw in plus, excitation wavelength: 532 nm). The ex situ X-ray absorption spectra (XAS) including XANES and EXAFS of the samples (7100-7170 eV were collected at the Spring-8 14b2 (Japan), where a pair of channel-cut Si (111) crystals was used in the monochromator. The storage ring was working at the energy of 8.0 GeV were average electron current of 99.5 mA. XAS data pre-treatment was carried out by means of ATHENA graphical utility Data reduction, data analysis, and EXAFS fitting were performed and analyzed with the Athena and Artemis programs of the Demeter data analysis packages[53] that utilizes the FEFF6 program[54] to fit the EXAFS data. The energy calibration of the sample was conducted through Fe foil, which as a reference was simultaneously measured. A linear function was subtracted from the pre-edge region, then the edge jump was normalized using Athena software. The $\chi(k)$ data were isolated by subtracting a smooth, third-order polynomial approximating the absorption background of an isolated atom. The $k^3$-weighted $\chi(k)$ data were Fourier transformed after applying a HanFeng window function ($\Delta k = 1.0$). For EXAFS modeling, the global amplitude EXAFS ($CN, R, \sigma^2$ and $\Delta E_0$) were obtained by nonlinear fitting, with least-squares refinement, of the EXAFS equation to the Fourier-transformed data in $R$-space, using Artemis software, EXAFS of the Fe foil are fitted and the obtained amplitude reduction factor $S_0^2$ value (0.811) was set in the EXAFS analysis to determine the coordination numbers ($CNs$) in the Fe-N/C/M scattering path in sample.

### Preparation of photoelectrode

Titanium dioxide ($TiO_2$) photoanode was prepared by modified hydrothermal method. In a typical procedure, 7.5 mL of deionized water was mixed with 7.5 mL of concentrated hydrochloric acid (37 wt.%, Titan, China), the mixture was stirred for 10 min at 25 ℃. Then, 0.3 mL of titanium butoxide (Sinopharm Chemical ReagentCo., Ltd, China) was added into the mixture again. After stirring for another 10 min, the mixture was transferred to a 50 mL Teflon-lined stainless autoclave (Chem$^N$, China). Then, a piece ($2.5 \times 1.5 \times 0.1 cm^3$) of the cleaned transparent conductive fluorine-doped tin oxide (FTO, 10 Ohm, Foshan Shiyuanjing Glass CO., China) substrate was cleaned by ultrasonication in a mixed solution of 2-propanol (Titan, China), deionized-water, and acetone (Titan, China) with volume ratios of 1:1:1

for 20 min. The FTO substrate was placed into the autoclave, and heated at 170 °C for 5 h. After reaction, the film was taken out and washed with deionized water. After dried up in air, the $TiO_2$ film on FTO substrate was annealed at 550 °C for 3 h in air. The as-prepared $TiO_2$ film on FTO substrate were obtained as the $TiO_2$ electrode. $BiVO_4$ photoanode was synthesized using a coprecipitation method as previously reported[55]. Firstly, BiOI film was electrodeposited on FTO glass (4 cm × 2 cm × 0.2 cm) after ultrasonic cleaning with acetone, isopropanol and water, respectively. A total of 491 mL of 0.4 M KI aqueous solution was mixed with 920 uL of concentrated nitric acid and 9 mL of water at pH = 1.6, and then 0.04 M of Bi $(NO_3)_3$·$5H_2O$ was added to obtain a transparent and clear solution. Then 50 mL KI/Bi $(NO_3)_3$ solution was taken, 0.3623 g (65.7 mM) p-benzoquinone was added, stirred again for 20 min, and filtered with an aqueous filter membrane (0.2 μm) and a needle tube. BiOI thin films were prepared in a three-electrode system of saturated calomel, Pt electrode and FTO, and −0.144 V (vs. SCE) bias electrodeposition for 120 s. Then, 0.2 M VO(acac)$_2$ DMSO solution was sonicated for 15 min to obtain a transparent and clear solution. The DMSO solution was dropped onto the prepared BiOI film at 55 uL cm$^{-2}$, placed flat in a rectangular quartz boat without a lid, placed in a muffle furnace at a heating rate of 2 °C min$^{-1}$ to 450 °C and maintained for 2 h, then cooled naturally. Finally, the obtained electrode was slowly stirred and immersed in 1.0 M KOH for 15 min to remove the by-product $V_2O_5$ impurities on the electrode surface, and the $BiVO_4$ electrode was obtained.

## Preparation of PBA electrode

FeFe PB, CoFe PBA, NiFe PBA and CuFe PBA was synthesized by a simple coprecipitation method.: In a typical FeFe-PB synthesis, 110 mg of K$_4$[Fe(CN)$_6$]·3H$_2$O (Aladdin, China) and 3.8 g of polyvinylpyrrolidone K30 (PVP, Titan, China) were added into a HCl solution (0.1 mol L$^{-1}$, 40 mL) under magnetic stirring. After 30 min, a clear solution was obtained and transferred into a vial. Then, the vial was sealed and placed in an electric oven and heated at 80 °C for 12 h. After aging, the precipitates were collected by centrifugation, and washed in de-ionized water and ethanol for six times. After drying at 25 °C for 12 h, the PB crystals were obtained. CuFe PBA powder was synthesized using a coprecipitation method. Briefly, 100 mL of 0.05 M Cu(NO$_3$)$_2$ and 100 mL of 0.05 M K$_3$[Fe(CN)$_6$] were simultaneously added to 500 mL of deionized water slowly, stirred fiercely for 30 min, and then placed in a fume hood for 24 h. The precipitate was washed several times with deionized water and ethanol, collected by centrifuge and dried in a vacuum oven at 60 °C. NiFe PBA: Firstly, 1300 mg K$_3$[Fe(CN)$_6$]was dissolved in 200 mL deionized water in A bottle. Then, 1400 mg NiCl$_2$ and 3.300 mg trisodium citrate were dissolved in a B flask containing 200 mL deionized water. Mix A and B bottle solution and let stand. After 24 h, the yellow precipitate was centrifuged, washed, and dried. CoFe PBA: Firstly, 50 mL deionized water was measured and 357 mg of Na$_4$[Fe(CN)$_6$] was ultrasonically dissolved in a vial. Then, 50 mL of deionized water was taken and 357 mg of CoCl$_2$ and 2500 mg of trisodium citrate were dissolved in a B bottle. Next, the A bottle solution and the B bottle solution were evenly mixed and stood at room temperature for 24 h. PBAs powder was used to further prepare electrodes. The PBAs electrode was fabricated by mixing the as-prepared PBAs powder, carbon black (KJ Group, China), and polyvinylidene fluoride (PVDF, HSV900, KJ Group, China) in N-methyl-2-pyrrolidone (NMP, Aladdin, China) at a weight ratio of 7:2:1. The slurry was coated onto carbon cloth (1 × 2 cm$^2$) and dried under vacuum at 100 °C for 24 h to obtain the PBAs cathode.

## Pollutant degradation methods

For the IAPEC system, $TiO_2$ or $BiVO_4$ electrode as the photoanode, and PBA electrode was used to as counter electrode to form a two-electrode system. Simulated wastewater is prepared by deionized water with different concentrations of NaCl with different ppm of pollutants.

Real wastewater was collected at a coal chemical plant in Inner Mongolia, China (42.46°,113.12°). Seawater was collected at Huludao, Liaoning (40.70°, 121.03°), Dongtou, Zhejiang, (27.85°, 121.153°), and Zhangzhou, Fujian (24.50°, 117.71°). The photoanode and counter electrode were immersed in 60 mL wastewater with magnetic stirring, a Xenon lamp (CEL-HXF300, Beijing Aulight Co., China) with an air mass filter AM1.5 and an irradiation of 100 mW cm$^{-2}$ was used to irradiate the photoanode. The I-t curve measurements were performed on a CHI electrochemical workstation (CHI660E, CHI Version 14.01) by recorded the changes of voltage and current during the degradation of pollutants. Take samples to test the degradation performance of contaminants.

We compared the degradation rates of target contaminants in tradition PEC (with irradiation, with $TiO_2$ as photoanode and platinum sheet electrode (1*1 cm$^2$) as the counter electrode, the applied bias is 0, 0.5 and 1 V respectively.), PC (irradiation only, with $TiO_2$ photoanode as photocatalyst), electrochemical (the applied bias (1 V) only)[28]. The controls of all experiments were carried out in the same concentration of sodium chloride as IAPEC. To enhance comprehension of the experimental conditions for PC, EC, and PEC, a schematic diagram is presented for reference. (Supplementary Fig. 50).

## Data fitting processing

A pseudo-first-order kinetic model was used to fit the experimental data; the model was expressed as Eq. 1

$$\ln\left(C_t/C_0\right) = -kt \qquad (1)$$

Where $C_O$ is the initial concentration of MB. $C_t$ is the concentration of MB at time $t$, and $k$ is the pseudo first-order rate constant.

## Sustainability evaluation

The energy consumption requirements in the degradation process of the system are analyzed by calculating the electricity consumed by the degradation of 1 g pollutants. The calculation formula is as Eq. 2:

$$W = (UI + P)\,h/1000 \times x \qquad (2)$$

Where $W$ is Energy consumption (kwh g$^{-1}$ MB), $U$ is power supply voltage (V), $I$ is power supply current (A), $h$ is reaction time, $P$ is simulation lamp power (mW cm$^{-2}$), $x$ is MB reduction value (g L$^{-1}$).

Carbon dioxide emission of household electricity (kg) = power consumption (kwh) x 0.785.

Note: The average industrial electricity consumption in Shanghai is 0.725 yuan per KWh, and the exchange rate between RMB and US dollar is 1 \$ = 6.9050 ¥ .

## The theoretical Computational methods of the Gibbs free energy for different reactions

The Vienna Ab-initio Simulation Package was employed to conduct all the DFT calculations in this progress[56,57]. The Perdew-Burke-Ernzerhof (PBE) functional, a generalized gradient approximation (GGA) method, was utilized to describe the exchange-correlation effects[58,59]. The projected augmented wave (PAW) method was employed to account for core-valence interactions[60].

For the expansion of wave functions, a plane wave cutoff energy of 400 eV was selected. Structural optimization was carried out until the energy convergence reached $1.0 \times 10^{-5}$ eV and the force convergence were at 0.02 eV Å$^{-1}$.

In the case of the different PBAs system, the Brillouin zone was sampled using a 3 × 3 × 3 grid centered at the gamma ($\Gamma$) point. For Pt (110), a vacuum space of 20 Å was incorporated above the surfaces to prevent periodic interactions. The Brillouin zone sampling for Pt (110) involved a 3 × 3 × 1 grid centered at the gammapoint. To accurately capture dispersion interactions, Grimme's DFT-D3 methodology was applied[61].

## Computational parameters of the degradation mechanism and paths

Here, Gaussian 16 combined with B3LYP-D3BJ/TZVP/SMD (water) level is used for configuration optimization and frequency calculation based on density functional theory. The fukui functions including $f^+$, $f^-$, $f^0$ and dual descriptor were also calculated by finite difference of electron densities of neutral, anionic, and cationic state of the molecule using Multiwfn (Multiwfn3.8) which is a powerful wave function analysis software[62].

## Data availability

The data that supports the findings of the study are included in the main text and supplementary information files. Raw data are available from the corresponding authors upon request.

## Code availability

No custom code or mathematical algorithm was used.

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

## Acknowledgements

The authors acknowledge the financial support by the National Key Research and Development Program of China (No. 2023YFE0122500), National Natural Science Foundation of China (No. 42107226, 42377359, 42277401) and Open Project of State Key Laboratory of Urban Water Resource and Environment, Harbin Institute of Technology (No. ES202223).

## Author contributions

In this manuscript, Q.D., Z.L., and L.T. conceived the idea, planned the study, designed the experiment, analyzed the data, and composed the manuscript. Q.D. and W.Z. performed all of the experiments with the assistance of J.L., L.W., D-L.W., D-J.W., and L.T. coordinated and supervised the project. All of the authors have given approval to the final version of the manuscript.

## Competing interests

The authors declare no competing interests.
