## [Peer Review File · Nature Communications]

Bias-Free Driven Ion- Assisted Photoelectrochemical System for Sustainable Wastewater TreatmentREVIEWER COMMENTS

Reviewer #1 (Remarks to the Author):

This study from Tang et.al reported a bias-free driven ion assisted photoelectrochemical (IAPEC) system for pollutant degradation. The authors claimed that a new ion-electron process can promote the electron-hole separation at photoanode and meanwhile the anion such as oxidation of Cl⁻ are used to generate radicals, which promotes the pollutant degradation as a result. The results seems interesting, and I would recommend it for publication after the following questions have been addressed.

1. There is a concern about the new concept "ion-electron coupling process". As protons are also ions, does the so called "ion-electron coupling process" can be distinguished from proton coupled electron process?
2. Following the above question, why does the Prussian blue electrodes do not take proton but only cation ions? More explanation is needed.
3. Fig.2b. Does it really make sense to compare the energy profile of HER over Pt and ion insertion over PBA? The materials properties of Pt and PBA are different. I would except to see what the energy profile for proton ion insertion over PBA is.
4. Typos. The authors put the wrong reference of Figure numbers for all figures in the part "Investigation of IAPEC system degradation performance for saline wastewater". e.g., Line 183 Fig.3a should be Fig.4a.
5. I have a question how does the author setup the PEC performance test? What are the electrode and electrolyte. This is important when using it as a control group to compare its performance with the IAPEC.
6. Why does sulphate show lower activity, when comparing with chloride?
7. The total organic content (TOC) needs to be done to support the degradation of pollutants.
8. The selected 3 compounds MB, IBP and CBZ are simple chemical molecules. Does the system work in other pollutants? Particularly some Refractory chemicals.
9. Will the decomposed species of MB be more toxic, as chloride radicals were produced in the process.

Reviewer #2 (Remarks to the Author):

General Comment:

"This manuscript reports an ion-assisted photoelectrochemical (IAPEC) system using PBA as the electrode material. The authors claimed that as a robust and reversible photoelectron-cation acceptor, PBA can produce strong oxidant free radicals. Consequently, the IAPEC system, consisting of TiO₂ and PBA, achieved high levels of pollutant degradation using seawater as an additive. However, the claims described are insufficiently supported. Therefore, the present manuscript is not sufficiently significant to be published in Nature Communications."

1. In Fig. 2a, the author showed photocurrent density over 15 mA cm^{-2} using the IAPEC system. However, it is impossible with TiO_2 photocurrent because the maximum photocurrent density of TiO_2 cannot be above 3 mA cm^{-2} under solar simulated light (100 mW cm^{-2}) due to its large band gap over 3.0 eV . The author should provide a more detailed explanation of how they achieved such a high photocurrent density.

2. The authors compared the HER on Pt (110) surface and cation insertion into PBA for the comparison of typical PEC and IAPEC. However, HER and cation insertion are clearly different reactions (mechanisms, reduction potential of reactions, etc.), so the comparison is not valid. To demonstrate the superiority of IAPEC, a comparison of identical reactions is necessary.

3. The authors measured the OPV of the IAPEC system using different types of PBAs as cathode and the typical PEC system and confirmed the higher voltages of the IAPEC system. However, no reason is given for the use of CuFe-PBA rather than CoFe-PBA, which has the highest voltage (1.19V). Also, the Gibbs free energies have been calculated when different cations are inserted into the PBA. However, the PBAs with different metals have different lattice sizes, so they have different energies when ions are inserted. In this manuscript, only calculations for CuFe-PBA were performed.

4. The points in Fig. 3c are from Table 1, not Table 2. I recommend combining Table 1 and 2 to avoid confusing the reader. Also, the notations for Fig. 4 are all incorrectly labeled as Fig. 3. They should be corrected. Please check the footnote in Fig. 4b. Its graph can't explain the results about the three pollutants.

5. I have some questions: Why is there no analysis of the Fe-M bond after Na insertion? When a cation is inserted in PBA, the cubic structure changes to a rhombohedral structure, but there is no analysis of this.

6. The authors claimed that the XRD patterns after the insertion of Na^+ ion showed large angles skewed. However, in Supplementary Fig. 9, the change in large angles is unclear, not only for Na^+ ion but also for other cations.

7. The experimental conditions about PC, EC, and PEC are unclear: Whether the experiment was conducted in sulfate or chloride. Whether it was bias-free or applied bias. If bias was applied, under what conditions.

8. In Supplementary Fig. 23, the photocurrent density in the electrolyte of NaCl and KCl does not correspond to the Gibbs energy calculation results in Fig. 2b. I think the authors have to provide more evidence or reasons for these issues.

9. No error bars or standard deviations were reported for any measurements. The authors must reproduce all of their measurements three times in total with independently prepared electrodes and should report the data with averages and standard deviations. All activity should also be reported with standard errors based on appropriate error propagation using the standard deviations from the reproduced measurements. This will verify to the readers that these are reproducible results and not one-off measurements.

10. The decrease in pH after the degradation of MB indicates an increase in H⁺ ions (or a decrease in OH⁻ ions). I think the authors have to provide more evidence or reasons for these issues."

Point-by-point response to the reviewers' comments (in blue)

Referee #1:

General comments: This study from Tang et.al reported a bias-free driven ion assisted photoelectrochemical (IAPEC) system for pollutant degradation. The authors claimed that a new ion-electron process can promote the electron-hole separation at photoanode and meanwhile the anion such as oxidation of Cl^- are used to generate radicals, which promotes the pollutant degradation as a result. The results seem interesting, and I would recommend it for publication after the following questions have been addressed.

Response: We appreciate reviewer's valuable time and efforts in reviewing our manuscript, and acknowledge that the results we presented were interesting. In order to consolidate the observed results and proposed concept, we have addressed the reviewers' questions accordingly.

Comment 1: There is a concern about the new concept "ion-electron coupling process". As protons are also ions, does the so called "ion-electron coupling process" can be distinguished from proton coupled electron process?

Response: We appreciate the insightful comment regarding the distinction between the "ion-electron coupling process" and the specific case of the "proton-coupled electron process". The reviewer correctly points out that protons, as ions themselves, fall within the broader category of ions. We agree that it is essential to clarify and differentiate our concept from a more generalized "proton-coupled electron process." In our study, we introduced the concept of the electron-ion acceptor cathode, highlighting its exceptional capability for fast ion-electron coupling. This characteristic plays a pivotal role in enhancing the separation of electrons and holes at the photoanode, constituting a key innovation in our work. To address the valid distinction between our concept and the broader understanding of "proton-coupled electron processes," we have made a deliberate effort to define our concept more explicitly as the "**inserted ion-assisted electron coupling process.**" By doing so, we aim to emphasize the critical role of inserted ions in driving the photoelectrochemical reactions, distinguishing it from processes primarily reliant on protons.

Based on above discussion, we revised the Manuscript as follows:

Line 79-81: "This system incorporates inorganic electron ion receptor materials capable of

reversible electron-ion storage as cathodes, enabling an **electron-inserted ion** coupling route for efficient transfer of photogenerated electrons.”

Revised Fig. 1 | Schematic diagram of the mechanism for of photogenerated carriers' separation pathway and the electron transfer and storage pathways behind different system. (a) Traditional PEC system. (b) **Our designed bias-free driven inserted ion-assisted photoelectrochemical (IAPEC) system.**

Comment 2: Following the above question, why does the Prussian blue electrodes do not take proton but only cation ions? More explanation is needed.

Response: Thank you for highlighting the importance of elucidating the selective cationic preference of Prussian blue electrodes (PBA) over protons. We will elaborate on this issue carefully from the following two aspects:

1. PBA material functions as a container for both electrons and ions. As mentioned by the reviewer, it is crucial to recognize its selectivity difference for various ion insertions. Therefore, we conducted theoretical calculations on the free energies for the insertion of four ions (Na^+ , K^+ , NH_4^+ , and H^+ ions) in different PBA materials, including CuFe PBA and three others (FeFe PB,

CoFe PBA, and NiFe PBA) mentioned in this work (**Table R1**). These results demonstrate that the free energy required for the insertion of H⁺ ions into the PBA material is significantly higher compared to the other three cations (**Fig.2b, Supplementary Fig. 10-12**). This strongly suggests that, in theory, PBAs have a higher tendency to selectively accommodate the three cations rather than H⁺ ions.

2. Considering practical aspects related to the electrolyte solution used in our wastewater treatment system, it's important to note that the electrolyte solution typically doesn't contain high concentrations of H⁺ ions. To further clarify this point, we utilized the average pH of seawater (pH= 8.3) to calculate the corresponding H⁺ ions concentration ($[H^+] = 5 \times 10^{-9} \text{ mol L}^{-1}$). Considering the average concentration of NaCl in seawater, the corresponding Na⁺ ions concentration ($[Na^+] = 0.55 \text{ mol L}^{-1}$), the substantial ion concentration difference makes Na⁺ ions the primary insertion ions.

Table R1 Theoretical calculations PBAs insertion free energy of four ions.

Ions \ PBAs	Ions			
	Na ⁺	K ⁺	NH ₄ ⁺	H ⁺
FeFe PB (eV)	-34.54	-40.43	-35.59	-1.11
CoFe PBA(eV)	-24.75	-29.33	-26.18	-0.55
NiFe PBA(eV)	-28.39	-33.51	-25.41	-0.71
CuFe PBA(eV)	-29.64	-34.83	-30.84	-1.13

Revised Fig. 2b The theoretical calculation of the Gibbs free energy change for the CuFe PBA structure during the insertion process of Na⁺, K⁺, NH₄⁺ and H⁺ ions.

Supplementary Figure 10 | The theoretical calculation of the Gibbs free energy change for the FeFe PB structure during the insertion process of Na⁺, K⁺, NH₄⁺ and H⁺ ions.

Supplementary Figure 11 | The theoretical calculation of the Gibbs free energy change for the NiFe PBA structure during the insertion process of Na⁺, K⁺, NH₄⁺ and H⁺ ions.

Supplementary Figure 12 | The theoretical calculation of the Gibbs free energy change for the CoFe PBA structure during the insertion process of Na⁺, K⁺, NH₄⁺ and H⁺ ions.

PBA structure during the insertion process of Na^+ , K^+ , NH_4^+ and H^+ ions.

Based on above discussion, we revised the Manuscript and supplementary as follows:

Line 136-143: “Furthermore, we conducted a comparative analysis of the free energies associated with different ion insertions into the PBA electrode (**Supplementary Fig. 10-12**). We observed a significant reduction in the free energy of the PBA structure after ion insertion. Notably, compared to H^+ ions, PBAs exhibited a stronger preference for the selective insertion of the three cations. As illustrated in **Fig. 2b**, the free energies for the insertion of Na^+ , K^+ and NH_4^+ ions by CuFe-PBA were measured at -34.54 , -40.43 , and -35.59 eV, respectively, while the free energy associated with inserting H^+ ions was notably higher at -1.11 eV.”

Comment 3: Fig.2b. Does it really make sense to compare the energy profile of HER over Pt and ion insertion over PBA? The materials properties of Pt and PBA are different. I would expect to see what the energy profile for proton ion insertion over PBA is.

Response: We appreciate the reviewer’s comment and their valuable suggestion.

As correctly pointed out, comparing the energy profiles of the Hydrogen Evolution Reaction (HER) on platinum (Pt) with ion insertion into Prussian Blue Analogs (PBA) may initially appear unconventional due to the differences in material properties.

Firstly, we would like to emphasize that our focus is on elucidating the mechanisms that contribute to the significant increase in the transfer rate of photogenerated electrons in our unbiased driven ion-assisted photoelectrochemical (IAPEC) systems. The rapid ion-electron coupling capabilities of the PBA electrode play a pivotal role in enhancing electron-hole separation. Consequently, electron-coupled ion insertion in PBA stands as the fundamental reaction in the IAPEC system, analogous to Pt-catalyzed HER reactions in traditional PEC systems. This justifies our choice of comparing the energy barrier of key reaction steps in these two systems.

While considering the reviewer's constructive suggestion, we also recognized the necessity of making comparisons under the same reaction principle. Therefore, we also added calculations of the energy barrier for H^+ ion insertion into PBAs. As detailed in **Table R2**, the free energy required for H^+ ion insertion by CuFe- PBA is -1.11 eV. Notably, this value is significantly higher than the energy barrier for the insertion of the other three cations, yet it remains lower than the energy barrier of 0.47 eV for the HER reaction. This result indicates that even in the insertion mode, protons are much

more challenging to insert compared to ions like sodium and potassium as discussed in this work.

Table R2 The free energy of H⁺ ions insertion into different PBA and Pt catalyzed HER reaction

	CuFe PBA	CoFe PBA	NiFe PBA	FeFe-PB	Pt (HER)
Free energy	-1.11 eV	-0.55 eV	-0.71 eV	-1.13 eV	-0.47 eV

Table R1 Comparison of H⁺ ions inserted into different PBAs and comparison of three cations insertion into PBAs.

PBAs \ Ions	Ions			
	Na ⁺	K ⁺	NH ₄ ⁺	H ⁺
FeFe PB(eV)	-34.54	-40.43	-35.59	-1.11
CoFe PBA(eV)	-24.75	-29.33	-26.18	-0.55
NiFe PBA(eV)	-28.39	-33.51	-25.41	-0.71
CuFe PBA(eV)	-29.64	-34.83	-30.84	-1.13

Comment 4: Typos. The authors put the wrong reference of Figure numbers for all figures in the part “Investigation of IAPEC system degradation performance for saline wastewater”. e.g., Line 183 Fig.3a should be Fig.4a.

Response: We thank the reviewer for this meticulous comment. We carefully checked and corrected the order of all the figures and tables.

Comment 5: I have a question how does the author setup the PEC performance test? What are the electrode and electrolyte. This is important when using it as a control group to compare its performance with the IAPEC.

Response: Thanks for your comment. According to your suggestion, we added the experimental details for the control group in the performance test of the PEC system. The test of IAPEC system and PEC system consists of electrochemical workstation, 100 ml quartz electrolyze, xenon lamp and magnetic stirrer. Under simulated solar illumination (intensity of 100 mW cm⁻²), 60 mL high salt organic wastewater was added to test the performance of different systems in *I-T* mode.

Electrode: The photoelectrode of PEC system uses the same TiO₂ electrode as the IAPEC system, the difference is that the electrode is the platinum sheet electrode (1*1 cm²) used in PEC system as the electrode, while the PBA electrode with the same active area is used in IAPEC system.

Electrolyte: The electrolyte of PEC system adopts NaCl solution or seawater spiked with corresponding concentration of pollutants, which is completely consistent with IAPEC system.

To ensure an accurate and replicable comparison with IAPEC performance, we have included a more detailed description of the experimental method in the Experimental section, along with specific experimental conditions. The relevant content is as follows:

Experimental section:

“Pollutant degradation methods.

For the IAPEC system, TiO₂ or BiVO₄ electrode as the photoanode, and PBA electrode was used to as counter electrode to form a two-electrode system. Simulated wastewater is prepared by deionized water with different concentrations of NaCl with different ppm of pollutants. Real wastewater was collected at a coal chemical plant in Inner Mongolia, China (42.46°,113.12°). Seawater was collected at Huludao, Liaoning (40.70°, 121.03°), Dongtou, Zhejiang, (27.85°, 121.153°), and Zhangzhou, Fujian (24.50°, 117.71°). The photoanode and counter electrode were immersed in 60 mL wastewater with magnetic stirring, a Xenon lamp (CEL-HXF300, Beijing Aulight Co., China) with an air mass filter AM1.5 and an irradiation of 100 mW cm⁻² was used to irradiate the photoanode. The *I-t* curve measurements were performed on a CHI electrochemical workstation (CHI660E, CHI Version 14.01) by recorded the changes of voltage and current during the degradation of pollutants. Take samples to test the degradation performance of contaminants.

We compared the degradation rates of target contaminants in tradition PEC (with irradiation, with TiO₂ as photoanode and platinum sheet electrode (1*1 cm²) as the counter electrode, the applied bias is 0, 0.5 and 1V respectively.), photocatalytic (PC, irradiation only, with TiO₂ photoanode as photocatalyst), electrochemical (EC, the applied bias (1 V) only)²⁸. The controls of all experiments were carried out in the same concentration of sodium chloride as IAPEC. To enhance comprehension of the experimental conditions for PC, EC, and PEC, a schematic diagram is presented for reference. (Supplementary Fig. 50).

Supplementary Figure 50 | A schematic diagram for the experimental conditions of IAPEC, PC, EC and PEC.

Comment 6: Why does sulphate show lower activity, when comparing with chloride?

Response: We appreciate the reviewer for raising this significant question, prompting us to elaborate on the varying activity of our IAPEC system in degrading pollutants in sulfate and chloride medium.

Initially, we characterized the types of active species generated by our IAPEC system in sulfate media using electron spin resonance (ESR). By utilizing DMPO as a trap for $\cdot\text{OH}$ and $\cdot\text{SO}_4^-$, characteristic peak signals of DMPO- $\cdot\text{OH}$ and DMPO- $\cdot\text{SO}_4^-$ appeared. Notably, even with the introduction of pollutants, the detected free radical signals remained relatively stable, indicating a consistent generation rate of $\cdot\text{OH}$ and $\cdot\text{SO}_4^-$ in our system (**Supplementary Fig. 39-40**) (*Chem Eng J.* **459**, 141474 (2023)). We further employed TEMP to validate the generation of h^+ . On the other hand, we used TEMP to confirm the generation of h^+ . **As shown in Supplementary Fig. 41**, the typical 1:1:1 triplet signal is in good agreement with the TEMP signal. After illumination, a significant decrease in the triplet signal was observed, confirming the high generation of h^+ in our system.

To discern the contributions of free radicals and direct oxidation, we conducted quenching experiments utilizing tertiary butanol (TBA) as a $\cdot\text{OH}$ scavenger and methanol (MeOH) as a $\cdot\text{OH}$ and $\cdot\text{SO}_4^-$ scavenger. **As shown in Supplementary Fig. 42**, the addition of quenchers modestly inhibited the removal of MB (Methylene Blue), showcasing a 7%~20.6% reduction. We also

calculated the contribution ratio of active species and direct oxidation based on the pseudo-first-order rate constants from the quenching experiment. The results indicated that direct oxidation (h^+) constituted the primary active species (58%), while $\cdot OH$ and $\cdot SO_4^-$ made up approximately 15% and 27% of the contributions, respectively, consistent with the ESR results.

However, the basic assumption of quenching experiments may not hold in certain reaction systems, leading to misinterpretation of the reaction mechanism. Therefore, we employed probe experiments and constructed a chemical kinetic model comprising direct oxidation and free radical oxidation to validate our above results. Specifically, oxalic acid (OA), nitrobenzene (NB), and phenol (PhOH) were used as probes for direct oxidation, $\cdot OH$, and $\cdot OH/\cdot SO_4^-$, respectively (*Appl Catal B-Environ.* **280**, 119418 (2021)). The results in **Supplementary Fig. 43** showed that the degradation rate of oxalic acid (0.03 min^{-1}) was about twice that of nitrobenzene (0.0164 min^{-1}) and phenol (0.0179 min^{-1}). The contribution ratio calculated by the chemical kinetic model confirmed the dominant role of h^+ (45%), while the quenching experiment underestimated the contributions of $\cdot OH$ (24.6%) and $\cdot SO_4^-$ (30.1%).

In a chlorine medium, the oxidation capacity of photogenerated holes suffices to oxidize chloride ions to $Cl\cdot$, and Cl^- loses electrons to form Cl_2 , which is further hydrolyzed to $HClO$. $Cl_2\cdot$ and $ClO\cdot$ are formed by the reaction of $Cl\cdot$ with Cl^- and ClO^- , respectively. Excess $HClO$ can also be decomposed into $Cl\cdot$ and $HO\cdot$ by absorbing light energy. The hydroxyl group can also react with free chlorine to form oxychloride free radical ($ClO\cdot$), and the signal of co-existence of chlorine-containing free radicals and hydroxyl group in the system was also proved by ESR test (**Fig. 5a**). After adding different radical quenchers to the reaction solution, $\cdot OH$ was quenched by adding NB, and the quasi-first-order reaction constant was reduced by 18.89%. However, after adding hole quenchers (EDTA-2Na) and different active chlorine substance quenchers, such as tert-butanol (TBA), $NaHCO_3$ and $Na_2S_2O_3$, the quasi-first-order reaction constants decreased by 91.60%, 90.35%, 86.31% and 88.77%, respectively, indicating that these chlorine-containing strong oxidizing groups played a crucial role in the degradation of pollutants. In summary, in the IAPEC system, the use of chloride salt as the electrolyte may produce more chlorine-containing groups and $OH\cdot$ co-degrade organic pollutants, while in the sulfate electrolyte, h^+ may mainly directly degrade organic pollutants (**Supplementary Fig. 44**). Different reaction mechanisms lead to lower degradation effect in the sulfate system than in the chloride salt system.

Supplementary Figure 39 | EPR spectra of DMPO-·OH and DMPO-·SO₄·- adducts in sulfate media in IAPEC system.

Supplementary Figure 40 | EPR spectra of DMPO-·OH and DMPO-·SO₄·- adducts in sulfate media with 10 ppm IBP in IAPEC system.

Supplementary Figure 41 | EPR spectra of TEMP-h⁺ adduct in sulfate media in IAPEC system.

Supplementary Figure 42 | The effect of quenchers on MB (10 ppm) degradation and the pseudo-first-order rate constants for quenching experiments in sulfate media in IAPEC system (inset picture). [TBA] and [MeOH] = 200 mM, [Na₂SO₄] = 0.5 M.

Supplementary Figure 43 Corresponding first-order kinetic curves of probe experiments for IAPEC system in sulfate media. [OA] = 100 mg/L, [Phenol] and [NB] as probe = 10 mg/L. [Na₂SO₄] = 0.5 M

Supplementary Figure 44 Comparison of contributions of active species calculated based on quenching and probe experiments (a) IAPEC under sulphate medium. (b) IAPEC under chloride medium.

Based on the obtained results and discussions, we have incorporated a discussion on the degradation mechanisms in sulfate media within the IAPEC system in the “Mechanism and degradation pathway insight into the IAPEC system” in the revised manuscript, as follows:

Line 449-458: “Meanwhile, we comprehensively analyzed the pathways for generating active

species within the IAPEC system in sulfate media, utilizing insights from electron paramagnetic resonance (EPR), quenching experiments, and molecular probe experiments (Supplementary Fig. 39-43)^{48, 49, 50}. As presented in Supplementary Fig. 43, the degradation rate of oxalic acid (0.03 min^{-1}) was notably approximately twice that of nitrobenzene (0.0164 min^{-1}) and phenol (0.0179 min^{-1}). Our chemical kinetic model calculations validated the prominent role of h^+ (45%), while the quenching experiment results underscored the contributions of $\cdot\text{OH}$ (24.6%) and $\cdot\text{SO}_4^-$ (30.1%). Furthermore, our calculations based on the chemical kinetic model emphasized the predominant role of chlorine free radicals ($\text{Cl}\cdot$, $\text{Cl}_2\cdot$, $\text{ClO}\cdot$) (71.46%), followed by $\cdot\text{OH}$ (18.89%) and free chlorine substances (ClO^-/HClO (8.4%)) (Supplementary Fig. 44).”

Comment 7: The total organic content (TOC) needs to be done to support the degradation of pollutants.

Response: Thanks for your comment. According to your suggestion, we conducted tests to measure the Total Organic Content (TOC) before and after treatment for all 8 pollutant models addressed in this study. The TOC data have been incorporated into the revised manuscript and are presented in Supplementary Fig. 24.

After 2 h treatment with the IAPEC system for various pollutants, the mineralization efficiency reached a maximum of ~50%. Notably, pollutants with inherently high organic carbon content, such as PFOA, also exhibited a mineralization efficiency of about 40%. These findings indicated that the IAPEC system not only demonstrates high pollutant remediation efficiency but also in its potential to avoid secondary pollution following pollutant degradation.

Supplementary Figure 24| The removal rate of TOC of eight model pollutants by IAPEC system

after 2 h reaction. (Experimental conditions: 60 mL 0.1 M NaCl solution, initial pH = 6.0 ± 0.1 , simulated solar light illumination 100 mW cm^{-2} , pollutant concentration: 10 ppm). Error bars representing the standard deviation of three replicate measurements.

Relevant discussions have been incorporated into the revised manuscript for the convenience of the reviewer, as listed below:

Line 263-269: “We assessed the Total Organic Content (TOC) of pollutants treated by the IAPEC system (**Supplementary Fig. 24**). The findings revealed that the IAPEC system achieves an impressive mineralization efficiency, reaching up to ~50%. Particularly significant is the observation that pollutants with higher organic carbon content, such as PFOA, also achieved a commendable mineralization efficiency of about 40%. These results emphasize that the IAPEC system excels not only in efficiently remediating pollutants but also in its potential to mitigate secondary pollution following pollutant degradation.”

Comment 8: The selected 3 compounds MB, IBP and CBZ are simple chemical molecules. Does the system work in other pollutants? Particularly some Refractory chemicals.

Response: Thank you for the review’s valuable advice. We acknowledge the importance of assessing the IAPEC system’s effectiveness on a broader range of pollutants, especially Refractory chemicals. Based on your suggestions, we expanded our pollution model to include data 5 typical refractory chemicals (**a total of 8 pollutants in this work**), which are:

1. **Bisphenol A (BPA):** An organic synthetic chemical primarily used in plastic products and epoxy resins production.
2. **4-Chlorophenol (4-CP):** A chlorinated phenolic compound commonly used as a germicide, solvent, and intermediate.
3. **Perfluorooctanoic Acid (PFOA):** A fluorinated compound previously utilized in the manufacturing of industrial and consumer products.
4. **Sulfamethoxazole (SMX):** A broad-spectrum antibiotic belonging to the sulfonamide class, often found in pharmaceutical wastewater.
5. **Cellulose Acetate Propionate (CAP):** A plastic material widely used in eyeglass frames and various industrial applications.

We evaluated the degradation efficiency of the above newly introduced pollutant models, namely Bisphenol A (BPA), 4-Chlorophenol (4-CP), Perfluorooctanoic Acid (PFOA), Sulfamethoxazole (SMX), and Cellulose Acetate Propionate (CAP) in both PEC and IAPEC systems (**Supplementary Fig.23**). After a 60 min treatment in the PEC system, the degradation rate of other pollutants was basically ignored except for 50% degradation of CAP. In contrast, in the IAPEC system, except for PFOA, all other pollutants reached ~80% degradation after 1h treatment. These results emphasize the IAPEC system's enhanced effectiveness in degrading a broader range of pollutants, especially Refractory chemicals.

Furthermore, we conducted a comparative analysis of the degradation effects on the eight pollutant models in three distinct seawater backgrounds following a 1 h treatment (**Fig. 5**). The outcomes demonstrate substantial degradation, exceeding 90% within just 20 min for all pollutants, except CAP and PFOA. CAP degradation was nearly complete within 1 h, while the degradation rate for the highly stable PFOA within the same timeframe ranged approximately from 65%. The enhanced degradation effect in authentic seawater may be attributed to the higher concentrations of chloride and sodium ions, considering that our sodium chloride concentration is 0.1 M, whereas the sodium chloride concentration in seawater is close to 0.55 M.

Supplementary Figure 23 | The degradations performances of PEC (a) and IAPEC (b) systems for Bisphenol A (BPA), 4-Chlorophenol (4-CP), Perfluorooctanoic Acid (PFOA), Sulfamethoxazole (SMX) and Cellulose Acetate Propionate (CAP). (Experimental conditions: 60 mL 0.1 M NaCl solution, initial pH = 6.0 ± 0.1, simulated solar light illumination 100 mWcm⁻², pollutant concentration: 10 ppm). Error bars representing the standard deviation of three replicate measurements.

Revised Figure 5e | The degradation 3D line diagram of three pollutants by the IAPEC system in eight different seawater backgrounds. (Pollutant concentration: 10 ppm, the electrolytes are taken from real seawater in Liaoning, Zhejiang and Fujian). Error bars representing the standard deviation of three replicate measurements.

Relevant discussions have been incorporated into the revised manuscript for the convenience of the reviewer, as listed below:

Line 256-262: “We further targeted a broad range of recalcitrant chemicals with our pollutant model, including Bisphenol A (BPA), 4-Chlorophenol (4-CP), Perfluorooctanoic Acid (PFOA), Sulfamethoxazole (SMX), and Cellulose Acetate Propionate (CAP), as shown in **Supplementary Fig.23**. After a 60 min treatment in the PEC system, the degradation rate of other pollutants was basically ignored except for 50% degradation of CAP. In contrast, in the IAPEC system, except for PFOA, all other pollutants reached 80% degradation after 1 h treatment.”

Line 388-396: “In light of this, we selected authentic seawater samples from three different regions of China (**Fig. 5d**) and spiked them with MB, IBP, CBZ, BPA, 4-CP, SMX, PFOA and CAP at a concentration of 10 ppm as target pollutants. No additional electrolyte was introduced, and neither pH adjustments nor further treatment procedures were undertaken. Based on the data presented in **Fig. 5e**, the degradation of the eight pollutants was observed to be completed within

30 min across all three types of seawater backgrounds, except for CAP and PFOA. Remarkably, CAP degradation was nearly complete within 1 h, while the degradation rate for the highly stable PFOA within the same timeframe ranged approximately around 65%.”

Comment 9: Will the decomposed species of MB be more toxic, as chloride radicals were produced in the process.

Response: We sincerely appreciate the reviewer for highlighting this concern.

Indeed, it is well-established that PEC-Cl systems, including IAPEC, can generate toxic chlorinated byproducts, such as ClO_3^- with a health limit of 0.7 mg L^{-1} , and ClO_4^- with a health limit of $1 \text{ }\mu\text{g L}^{-1}$ under full-spectrum solar radiation (European Food Safety Authority) (*Environ. Sci. Technol.* 2018, 52, 6317–6325). To address this concern, we conducted IC (Ion chromatography) tests to assess the levels of chlorates ClO_3^- and ClO_4^- in the IAPEC system after the degradation of MB, as illustrated in the revised **Supplementary Fig. 25**. The results indicate a ClO_3^- concentration of approximately 0.5 mg L^{-1} , with undetectable levels of ClO_4^- . Compared to recent PEC-Cl systems reported in the literature, IAPEC exhibits excellent pollutant degradation while displaying a superior inhibitory effect on the generation of toxic byproducts (**Supplementary Table 3**).

Furthermore, mechanistic studies have revealed that the formation of active chlorine species and toxic oxychloride byproducts from $\text{Cl}\cdot$ are valence-band dependent. The constrained formation of $\text{HO}\cdot$ and non- $\text{HO}\cdot$ -mediated reactive chlorine species formation synergistically suppresses ClO_3^- in PEC systems using BiVO_4 photoelectrode. (*Appl. Catal. B* 2021, 296, 120387). Hence, we also investigated the production of chlorates in the IAPEC system after the completion of MB degradation using a BiVO_4 photoanode. Our results demonstrated that BiVO_4 effectively removes pollutants, resulting in ClO_3^- levels less than 0.5 mg L^{-1} and undetectable ClO_4^- .

Besides, we applied LC-MS to analyze possible intermediates in the process of MB degradation. Some major intermediates were observed and their possible molecular structures were shown in **Supplementary Fig. S45-49**. Typically, these products were produced through a series of processes such as oxidation, ring opening and fragmentation. Ideally, these products would eventually mineralize into CO_2 and H_2O . The possible degradation pathways of MB in the PEC-Cl system were illustrated in **Fig. 6d**

Supplementary Figure 25 Concentrations of toxic oxychlorides during the IAPEC treatment of saline sewage with different Photoanode. (N.D. is not detected, experimental conditions: 60 mL 50 mM NaCl simulated saline sewage, using TiO₂ or BiVO₄ as photoanode, conducted in triplicate) Error bars representing the standard deviation of three replicate measurements.

Supplementary Table 3. Comparison of the photoelectrochemical performance of various PEC-Cl.

References	Condition	Photoelectrode	Degradation Rate	Chlorate
Environ. Pollut. 2020, 267 115605	50 mM NaCl, 350 W Xe lamp, 1.36 V vs. RHE, pH=7	WO ₃ /BiVO ₄	30 ppm urea 97 %, 90 min	46.3 mg L ⁻¹
Environ. Sci. Technol. 2019, 53, 6945–6953	50 mM NaCl, 150 W xenon lamp, 1.7 V vs. Ag/AgCl, pH 5	Sb–SnO ₂ /WO ₃ (16 cm ²)	30 ppm ammonia- N 99.2 %, 90 min	26.7 mg L ⁻¹
Appl. Catal. B: Environ. 2021, 296, 120387	0.1 M NaCl, 300 W Xe lamp, AM 1.5, 1.2 V vs. RHE, pH = 3	TiO ₂	10ppm phenol, 99.9 %, 120min	2.3 mg L ⁻¹
J. Hazard. Mater. 2021, 402, 123725	300 mg L ⁻¹ NaCl, 350 W Xe lamp, pH = 9	WO ₃ /BiVO ₄ (2.5*5 cm ²)	10 ppm ammonia- N, 99.3 %, 120 min	18.3 mg L ⁻¹
Environ. Sci. Technol. 2019, 53, 9926–9936	50 mM NaCl, 300-W Xe arc lamp, 0.5 V vs. Ag/AgCl, pH = 4	WO ₃ film (1*2 cm ²)	50 μM 4-CP, 99.9% 30 min	150.3 mg L ⁻¹
J. Hazard. Mater. 2023, 443, 130363	50mM NaCl, 300 W Xe lamp, AM 1.5, 0.5 V vs. SCE, pH =4	O _v -TiO ₂ (2*2 cm ²)	20 ppm 4-CP, 99%, 45min	1.0 mg L ⁻¹

This work (IAPEC system)	50 mM NaCl, 300 W Xe lamp, AM 1.5, pH =6	TiO ₂ (4.91 cm ²)	20 ppm MB 99%, 15 min	0.55 mg L ⁻¹
---	---	--------------------------	-------------------------

Relevant discussions have been incorporated into the revised manuscript and listed below:

Line 272-279: “Additionally, we monitored the concentration of the toxic byproduct chlorate during the degradation of MB in the IAPEC system using different photocathodes. As illustrated in **Supplementary Fig.25**, the results indicated that when employing a TiO₂ photocathode, the concentration of ClO₃⁻ was approximately 0.5 mg L⁻¹. In contrast, for the BiVO₄ system, the ClO₃⁻ concentration was around 0.15 mg L⁻¹ (with a health limit of 0.7 mg L⁻¹), and in both systems, the ClO₄⁻ concentration was undetectable²⁹. In contrast, IAPEC system demonstrates exceptional pollutant degradation while exhibiting a superior inhibitory effect on the generation of toxic byproducts (**Supplementary Table 3**).”

Referee #2:

General comments: This manuscript reports an ion-assisted photoelectrochemical (IAPEC) system using PBA as the electrode material. The authors claimed that as a robust and reversible photoelectron-cation acceptor, PBA can produce strong oxidant free radicals. Consequently, the IAPEC system, consisting of TiO₂ and PBA, achieved high levels of pollutant degradation using seawater as an additive. However, the claims described are insufficiently supported. Therefore, the present manuscript is not sufficiently significant to be published in Nature Communications.

Response: We are very grateful to the reviewer for the valuable time and effort that put into our manuscript. The proposed comments and suggestions are valuable and helpful for improving our manuscript. We carefully revised the manuscript and added more data to help enhance the robustness of our claims. We wish our revised manuscript can be better now.

Before replying to the specific comment of the reviewer, we would like to highlight the main revisions that we have made:

1. In response to your concern about the appropriateness of comparing the energy barrier of PBA-inserted ionic reactions with the energy barrier of Pt-catalyzed HER reactions, we respectfully agree with your comparison of the inadequacy of two different reaction energy barriers. In this case, as we proposed that the key step of electron transfer in IAPEC system lies in the reception of electrons by PBA, while the key of photogenerated electron transfer in PEC system lies in the HER reaction of H⁺ at the Pt electrode. Therefore, we choose to compare the key step energy barrier of photogenerated electron transfer in the two systems. Moreover, based on your suggestion, we added calculations of the energy barrier for PBA insertion of hydrogen ions. The results show that the energy barrier of PBA insertion into hydrogen ion is lower than that of HER reaction. The consistent results revealed that the free energy of the Prussian blue system after hydrogen ion insertion is significantly higher than that for other cat ions (Na⁺, K⁺ and NH₄⁺ ions). This indicates that, theoretically, PBAs tend to selectively accommodate these three cation ions over H⁺ ions. These results further support our work findings and enhance the relevance of the comparison.

According to your suggestion, we calculated the inserting energy for all PBAs with typical cations (Na⁺, K⁺, NH₄⁺ and H⁺) to support our claim of the significant advantage of PBA in enhancing electron-hole transfer to the electrode.

2. To further emphasize the superiority of our proposed system in wastewater treatment, we included tests for five additional refractory chemicals (a total of 8 pollutant models), including BPA (Bisphenol A), 4-CP (4-Chlorophenol), PFOA (Perfluorooctanoic Acid), SMX (Sulfamethoxazole) and CAP (Cellulose Acetate Propionate). We also tested TOC changes before and after degradation of pollutants by the IAPEC system, and evaluated the toxic byproduct chlorate. All these experiments have been statistically analyzed with error bars to thoroughly demonstrate the superior degradation performance of our system towards these pollutants.
3. To provide a clearer explanation of the mechanisms within our proposed system, such as the phase structure transition reactions during ion insertion in various PBA materials, we reanalyzed the XRD structures of the four PBA materials mentioned in this study before and after sodium ion insertion. Indeed, as mentioned by the reviewer, the crystal structure of PBA (FeFe PB, NiFe PBA, CoFe PBA) undergoes a transformation from cubic to rhombohedral structure after ion insertion. However, for CuFe PBA, there is no significant structural change. This is mainly due to the ultra-low strain structure of the CuFe PBA material prepared using the method in this study (proposed by Cui et al. in 2012, it was shown that this ultra-low strain structure of CuFe PBA maintains a highly stable structure, retaining 83% of its capacity even after 40,000 cycles of K^+ ions insertion and extraction). We also conducted synchrotron powder XRD tests and refined the structure to demonstrate the lattice changes in CuFe PBA crystal after Na^+ ions insertion.
4. We revised the implementation steps of the controlled experiments involved in the manuscript, and described in detail the specific differences between IAPEC and PEC, EC and PC experiments in the control group by means of a schematic diagram.

We believe these revisions have significantly improved the manuscript, addressing the concerns and suggestions raised by the reviewer. We hope that with these enhancements, our manuscript is now better suited for publication.

Comment 1: In Fig. 2a, the author showed photocurrent density over 15 mAcm^{-2} using the IAPEC system. However, it is impossible with TiO_2 photocurrent because the maximum photocurrent density of TiO_2 cannot be above 3 mAcm^{-2} under solar simulated light (100 mW cm^{-2}) due to its

large band gap over 3.0 eV. The author should provide a more detailed explanation of how they achieved such a high photocurrent density.

Response:

We are very grateful for your valuable comment. Our current density is calculated based on the electrode area of the cathode PBA. But, this comment has prompted us to recognize that such calculations can potentially result in misunderstandings. Therefore, we will use calculated current density based on area of the photoanode (TiO₂) in this work.

First, we carefully measured the light-receiving area of the photoanode, as shown in **Supplementary Fig. 7**, in the electrolytic cell used for our photochemical reactions. The window on the photoelectrode that receives light is approximately a circular area with a diameter of about 2.5 cm (The area approximately 4.91 cm²). This choice of area was determined based on the distribution of irradiance from our simulated light source, as depicted in **Supplementary Fig. 8**. It can be observed that the irradiance gradually decreases from the center to the periphery. Hence, within approximately 1.5 cm from the center, the irradiance from the simulated light source can be greater than 90 mW cm⁻². Subsequently, we conducted experiments by varying the light-receiving area on the photoelectrode to measure the corresponding photocurrent. As shown in **Fig. R1**, we observed a significant increase in current as the electrode size increased from 1 cm² to 5 cm². This indicates a clear relationship between current density and the area exposed to light.

Supplementary Figure 7 | Our IAPEC test unit (a) left view and (b) front view.

Supplementary Figure 8 | Distribution of the irradiation intensity of the simulated sunlight.

Figure R1 | The transient photocurrent values recorded for IAPEC system with and without sunlight irradiation in the electrolyte of 0.5 M NaCl, simulated solar light illumination 100 mW cm⁻² xenon lamp source at 60 s intervals.

Revised Fig. 2a The transient average photocurrent density diagram of traditional PEC and IAPEC under 100 mW cm⁻² intensity with different cation aqueous (0.5 M) solutions. Error bars representing the standard deviation of three replicate measurements.

Relevant discussions have been incorporated into the revised manuscript and listed below:

Line 118-121: “We utilized a modified hydrothermal method to synthesize an anatase phase titanium dioxide (TiO₂) electrode as photoanode for our designed PEC system. The detail process for all electrode material is illustrated with the relevant descriptions in **Supplementary Fig. 1-8** and the Experiment sections.”

Line 121-125: “The photocurrent densities generated by the IAPEC system (TiO₂-PBA) and traditional PEC system (TiO₂-Pt) were compared in different cationic electrolytes, as depicted in **Fig.2a**. The results revealed that the IAPEC system generated a photocurrent density of 2.96 mA cm⁻², which is over 30 times higher than that of the traditional PEC system (0.10 mA cm⁻²).”

Comment 2: The authors compared the HER on Pt (110) surface and cation insertion into PBA for the comparison of typical PEC and IAPEC. However, HER and cation insertion are clearly different reactions (mechanisms, reduction potential of reactions, etc.), so the comparison is not valid. To demonstrate the superiority of IAPEC, a comparison of identical reactions is necessary.

Response: We thank the reviewer for this comment and also the valuable suggestion.

As mentioned by the reviewers, we agree that comparing the energy profiles of the Hydrogen

Evolution Reaction (HER) over platinum (Pt) and ion insertion over Prussian Blue Analogs (PBA) may initially seem unconventional due to the differing material properties. However, in our work, we aimed to explore the energy landscape of critical reactions in Photoelectrochemical (PEC) systems. The comparison was made to evaluate the kinetic properties and energy barriers associated with these reactions, as the HER is often a key step in PEC processes. Platinum serves as a benchmark catalyst for the HER, and thus, its inclusion allowed us to assess the relative performance of PBA in facilitating these reactions.

While considering the reviewer's constructive suggestion, we acknowledged the importance of conducting comparisons based on the identical reactions. Thus, we also performed calculations the free energies of H^+ ion inserting into the PBA electrode. As detailed in **Table R1**, the free energy required for H^+ ion insertion by CuFe PBA is -1.11 eV. Notably, this value is significantly higher than the energy barrier for the insertion of the other three cations (**Table R2**), yet it remains lower than the energy barrier of 0.47 eV for the HER reaction. These results further support our work findings and enhance the relevance of the comparison.

Table R1 The free energy of H^+ ions insertion into different PBA and Pt catalyzed HER reaction

	CuFe PBA	CoFe PBA	NiFe PBA	FeFe-PB	Pt (HER)
Free energy	-1.11 eV	-0.55 eV	-0.71 eV	-1.13 eV	-0.47 eV

Table R2 Comparison of H^+ ions inserted into different PBAs and comparison of three cations insertion into PBAs.

Ions PBAs	Na ⁺	K ⁺	NH ₄ ⁺	H ⁺
	FeFe PB(eV)	-34.54	-40.43	-35.59
CoFe PBA(eV)	-24.75	-29.33	-26.18	-0.55
NiFe PBA(eV)	-28.39	-33.51	-25.41	-0.71
CuFe PBA(eV)	-29.64	-34.83	-30.84	-1.13

Comment 3: The authors measured the OPV of the IAPEC system using different types of PBAs as cathode and the typical PEC system and confirmed the higher voltages of the IAPEC system.

However, no reason is given for the use of CuFe-PBA rather than CoFe-PBA, which has the highest voltage (1.19V). Also, the Gibbs free energies have been calculated when different cations are inserted into the PBA. However, the PBAs with different metals have different lattice sizes, so they have different energies when ions are inserted. In this manuscript, only calculations for CuFe-PBA were performed.

Response: We appreciate the reviewer's insightful comments and provide further clarification.

Regarding the choice of CuFe PBA over CoFe PBA:

First, attributed to the simplicity and scalability of the CuFe PBA preparation method pioneered by Wessells et al. (*Nat. commons.* 2, 550, 2011). More importantly, the CuFe PBA prepared by this method showed excellent stability in ions insertion and extraction, and 83% of the original capacity could be retained after 40,000 discharge cycles. The excellent cyclic stability of the CuFe PBA was shown by XRD analysis to be due to the ultra-low strain of the structure during ion insertion and extraction (**This result was also confirmed in our XRD characterization results Supplementary Fig. 15-16**). This is also a key indicator for the selection and evaluation of electrode materials in the construction of a sustainable wastewater treatment system. In addition to the synthesis advantages, we considered the toxicity and cost implications. Copper is preferred due to its lower toxicity and cost-effectiveness compared to cobalt. Hence, in the context of sustainability and practical applicability, we chose CuFe PBA to investigate its performance in this work.

Regarding the Gibbs Free Energies for Ions Insertion:

To gain a better understanding of the ion insertion characteristics in PBAs with different metal coordination based on your comment, we have included calculations of the insertion energies for three other PBAs involving three different cations. **As depicted in the revised Supplementary Figure 10&12**, the inserting energies of CuFe PBA for the three ions are all lower than those of CoFe PBA.

Supplementary Figure 15 (a) XRD patterns of CuFe-PBA electrode before and after ion insertion in the IAPEC system (b) Enlarged view of the (200) crystal plane. (c) Enlarged view of the (220) crystal plane. (d) Enlarged view of the (400) crystal plane.

Supplementary Figure 16 (a) Schematic structure CuFe PBA sample before and (b) after Na⁺ ions insertion. (c) Rietveld refinement synchrotron PXRD pattern of CuFe PBA sample before and (d) after Na⁺ ions insertion.

Supplementary Figure 10 | The theoretical calculation of the Gibbs free energy change for the FeFe PB structure during the insertion process of Na⁺, K⁺, NH₄⁺ and H⁺ ions.

Supplementary Figure 12 | The theoretical calculation of the Gibbs free energy change for the CoFe PBA structure during the insertion process of Na⁺, K⁺, NH₄⁺ and H⁺ ions.

Comment 4: The points in Fig. 3c are from Table 1, not Table 2. I recommend combining Table 1 and 2 to avoid confusing the reader. Also, the notations for Fig. 4 are all incorrectly labeled as Fig. 3. They should be corrected. Please check the footnote in Fig. 4b. Its graph can't explain the results about the three pollutants.

Response: We thank the reviewer for this meticulous comment. We carefully checked and corrected the order of all the Figures and tables. We thank the reviewer for this meticulous comment. Thanks for your suggestion, we have combined Table 1 and Table 2 listed below. We also carefully checked and corrected the order of all the Figures and tables.

Supplementary Table 1 | Fe pre-edge peak analysis on ex situ samples and the average oxidation state of Fe species in samples.

Sample	pre-edge peak energy (eV)	E_0 (eV)	Average oxidation state
Fe foil	-	7112	+0
FeO	7112.43	7123	+2
Fe ₂ O ₃	7115.07	7128.1	+3
Before Na ⁺ ions insertion	7114.93	7127.6	+2.9
Before Na ⁺ ions insertion	7112.27	7123.4	+2.1

E_0 values obtained from the first derivation spectral, which is linearly related to the Fe valence state.

Fig. 4 | b, The degradation 3D line diagram of MB pollutants by IAPEC system under different PBA counter electrode. Error bars representing the standard deviation of three replicate measurements.

Comment 5: I have some questions: Why is there no analysis of the Fe-M bond after Na insertion? When a cation is inserted in PBA, the cubic structure changes to a rhombohedral structure, but there is no analysis of this.

Response: We thank the reviewer for providing a very insightful comment.

1. After the insertion of Na⁺ ions, the state of iron element does indeed undergo a change. In the CuFe PBA framework, iron (Fe) is coordinated by the CN group. Therefore, we use Raman spectra characterization to study the changes in the state of Fe. It can be seen that after Na⁺

ions insertion, the characteristic peak of the cyanide shifted in lower wavenumber peaks. Due to the frequency of the cyanide stretching vibration mode ($\nu(\text{CN})$) is sensitive to the surrounding chemical environment, so the cyanide coordinated with Fe^{II} shows a relatively lower wavenumber peaks than the cyanide coordinated with Fe^{III} (**Supplementary Fig. 17**).

2. As mentions by the reviewer, Prussian Blue Analogs (PBAs) undergo corresponding changes in its crystal structure during the process of the insertion of a cation. The changes in its structure are usually associated with the amount of ionic insertion, the ionic species, and the PBA species. The study of the crystal structural changes after the insertion of sodium ions is crucial for our proposed system mechanism research. Therefore, according to your suggestion, we conducted X-ray diffraction (XRD) characterization to assess the structural changes in all four PBAs electrode before and after Na^+ ion insertion in this work. As depicted in the revised **Fig. R2-4**, FeFe PB, CoFe PBA, and NiFe PBA exhibited a distinct transition from a cubic to a rhombohedral structure after Na^+ ions insertion. The crystal structure of CuFe PBA always maintains a face-centered cubic structure with the insertion of sodium ions, and the position of the diffraction peak shifts to a higher Angle of the diffraction peak, indicating the slightly shrink of framework. To further investigate the changes in the crystal structure of CuFe PBA before and after the insertion of sodium ions, we employed synchrotron powder X-ray diffraction to acquire high-resolution structural data, allowing for a refined analysis of the crystallographic information. **Supplementary Fig. 16** depicted the refined crystal structure before and after the insertion of sodium ions. CuFe PBA maintains its $\text{Fm}\bar{3}\text{m}$ structure after the insertion of sodium ions. And, with the insertion of sodium ions, the lattice parameter a changed from 10.110 to 10.076 Å. This can be attributed to superior stability of CuFe PBA, which has an ultra-low strain open frame structure (Nat. common. 2, 550, 2011).

Supplementary Figure 17| Raman spectra of CuFe PBA before and after Na⁺ ions insertion.

Figure R2| XRD patterns of FeFe PB electrode before and after Na⁺ ions insertion.

Figure R3 | XRD patterns of CoFe PBA electrode before and after Na⁺ ions insertion.

Figure R4 | XRD patterns of NiFe PBA electrode before and after Na⁺ ions insertion.

Supplementary Figure 15 | XRD patterns of CuFe PBA electrode before and after Na⁺, K⁺ and NH₄⁺ ions insertion. (b) Enlarged view of the (200) crystal plane. (c) Enlarged view of the (220) crystal plane. (d) Enlarged view of the (400) crystal plane.

Supplementary Figure 16 | (a) Schematic structure CuFe PBA sample before and (b) after Na⁺ ions insertion. (c) Rietveld refinement synchrotron PXRD pattern of CuFe PBA sample before and (d) after Na⁺ ions insertion.

Based on above results and discussion, we revised the Manuscript and Supplementary Information as follows:

Line 180-194: “The crystal structure of CuFe PBA always maintains a face-centered cubic structure with the embedding of sodium ions, and the position of the diffraction peak shifts to a higher Angle of the diffraction peak, indicating the slightly shrink of framework. To further

investigate the changes in the crystal structure of CuFe PBA before and after the insertion of sodium ions, we employed synchrotron powder X-ray diffraction to acquire high-resolution structural data, allowing for a refined analysis of the crystallographic information. **Supplementary Fig. 16** depicts the refined crystal structure before and after the insertion of sodium ions. CuFe PBA maintains its Fm3m structure after the insertion of sodium ions. And, with the insertion of sodium ions, the lattice parameter a changed from 10.110 to 10.076 Å. This can be attributed to superior stability of CuFe PBA, which has an ultra-low strain open frame structure²⁵. Raman spectra characterization was employed to study the changes in the state of Fe. It can be seen that after Na⁺ ions insertion, the characteristic peak of the cyanide shifted in lower wavenumber peaks. Due to the frequency of the cyanide stretching vibration mode ($\nu(\text{CN})$) is sensitive to the surrounding chemical environment, so the cyanide coordinated with Fe^{II} shows a relatively lower wavenumber peaks than the cyanide coordinated with Fe^{III} (**Supplementary Fig. 17**).²⁶

Comment 6: The authors claimed that the XRD patterns after the insertion of Na⁺ ion showed large angles skewed. However, in Supplementary Fig. 9, the change in large angles is unclear, not only for Na⁺ ion but also for other cations.

Response: Thanks for your kind reminder. To ensure a more precise and detailed representation of these changes, we have conducted further XRD measurements on the CuFe PBA electrodes. These measurements were performed at a slower scan rate or employing synchrotron powder X-ray diffraction to acquire high quality data. Subsequently, we have redrawn the XRD patterns, and the updated results presented in **Supplementary Fig. 15**. This revised figure clearly illustrated the shifts to larger angles for the peaks corresponding to the (200), (220), and (400) crystal planes after the insertion of ions. The data obtained from synchrotron powder X-ray diffraction also shifted towards higher angles.

Supplementary Figure 15 | XRD patterns of CuFe PBA electrode before and after Na^+ , K^+ and NH_4^+ ions insertion. (b) Enlarged view of the (200) crystal plane. (c) Enlarged view of the (220) crystal plane. (d) Enlarged view of the (400) crystal plane.

Comment 7: The experimental conditions about PC, EC, and PEC are unclear: Whether the experiment was conducted in sulfate or chloride. Whether it was bias-free or applied bias. If bias was applied, under what conditions.

Response: We apologize for any confusion regarding the experimental conditions.

We compared the degradation rates of target contaminants in tradition PEC (with irradiation, with TiO_2 as photoanode and platinum sheet electrode ($1 \times 1 \text{ cm}^2$) as the counter electrode, the applied bias is 0, 0.5 and 1V respectively.), photocatalytic (PC, irradiation only, with TiO_2 photoanode as photocatalyst), electrochemical (EC, the applied bias (1 V) only). (*Reference: Choi, W. et.al, In Situ Photoelectrochemical Chloride Activation Using a WO_3 Electrode for Oxidative Treatment with Simultaneous H_2 Evolution under Visible Light. *Environmental Science & Technology* 2019, 53, 9926-9936*). The controls of all experiments were carried out in the same concentration of sodium chloride as IAPEC. In order to give a more intuitive understanding of the experimental conditions of PC, EC and PEC, we have drawn a schematic diagram for your understanding (**Supplementary Fig. 50**.)

Additionally, in our previous manuscript, we compared the degradation effect of the PEC system without bias. In this revised version, we have included additional data illustrating the degradation effect of the PEC system on Methylene Blue (MB) under various applied bias pressures. This data

is presented in **Fig. 4a**. Interestingly, even with 1 V applied bias, the degradation effect of the PEC system on MB remained inferior to that of our IAPEC system without bias.

Supplementary Figure 50 | A schematic diagram for the experimental conditions of IAPEC, PC, EC and PEC.

Figure 4a | The degradation performance for MB by different system. Experimental conditions: 60 mL simulated saline sewage, initial pH = 6.0 ± 0.1, simulated solar light illumination 100 mW cm⁻², [MB]₀ = 10 ppm, chloride medium: [NaCl] = 0.1 mol L⁻¹, Sulfate: [Na₂SO₄] = 0.1 mol L⁻¹. All tests were performed in chlorine media, except for the IAPEC (sulfate) group. Error bands representing the standard deviation of three replicate measurements.

Based on above results and discussion, we revised the Manuscript and Supplementary Information as follows:

Experimental section:

“Pollutant degradation methods.

Line 639-657: “For the IAPEC system, TiO₂ or BiVO₄ electrode as the photoanode, and PBA electrode was used to as counter electrode to form a two-electrode system. Simulated wastewater is prepared by deionized water with different concentrations of NaCl with different ppm of pollutants. Real wastewater was collected at a coal chemical plant in Inner Mongolia, China (42.46°,113.12°). Seawater was collected at Huludao, Liaoning (40.70°, 121.03°), Dongtou, Zhejiang, (27.85°, 121.153°), and Zhangzhou, Fujian (24.50°, 117.71°). The photoanode and counter electrode were immersed in 60 mL wastewater with magnetic stirring, a Xenon lamp (CEL-HXF300, Beijing Aulight Co., China) with an air mass filter AM1.5 and an irradiation of 100 mW cm⁻² was used to irradiate the photoanode. The *I-t* curve measurements were performed on a CHI electrochemical workstation (CHI660E, CHI Version 14.01) by recorded the changes of voltage and current during the degradation of pollutants. Take samples to test the degradation performance of contaminants. We compared the degradation rates of target contaminants in tradition PEC (with irradiation, with TiO₂ as photoanode and platinum sheet electrode (1*1 cm²) as the counter electrode, the applied bias is 0 or 1V respectively.), photocatalytic (PC, irradiation only, with TiO₂ photoanode as photocatalyst), electrochemical (EC, the applied bias (1 V) only)²⁸. The controls of all experiments were carried out in the same concentration of sodium chloride as IAPEC. To enhance comprehension of the experimental conditions for PC, EC, and PEC, a schematic diagram is presented for reference. (Supplementary Fig. 50).”

Line XX: “First, we conducted a comparison of various systems in degrading methylene blue (MB, 10 ppm) under identical experimental conditions, including photocatalysis (PC), electrocatalysis (EC), traditional PEC (A bias voltage of 0 V or 1 V applied), and IAPEC in sulfate and chloride medium, respectively (Fig.4a). The results showed that IAPEC exhibited the best degradation performance in chloride medium, among the different systems^{27,28}. And the corresponding pseudo-first-order rate constant of IAPEC in chloride medium reached as high as 0.5 min⁻¹ in 60 min, compared to only 0.24, 0.012, 0.005, 0.004 and 0.003 min⁻¹ for IAPEC (sulfate), PEC(1V), PEC(0V), EC and PC respectively (Supplementary Fig. 20).”

Comment 8: In Supplementary Fig. 23, the photocurrent density in the electrolyte of NaCl and KCl does not correspond to the Gibbs energy calculation results in Fig. 2b. I think the authors have to provide more evidence or reasons for these issues.

Response: We appreciate your keen observation and your valuable comment. The Gibbs free energy of potassium and sodium ion insertion is quite close. Therefore, we conducted multiple experiments to eliminate the influence of accidental systematic errors. The obtained results indicate that the photocurrent in the potassium ion solution is slightly higher than that in the sodium ion solution, which is consistent with the calculated Gibbs free energy results. We have incorporated the new data with error bars into this manuscript. We updated the **Fig. 2a and Supplementary Figure 23 (New order is Supplementary Figure 34)** in the manuscript.

Revised Fig. 2a| The transient average photocurrent density diagram of traditional PEC and IAPEC under 100 mW cm⁻² intensity with different cation aqueous (0.5 M) solutions. Error bars representing the standard deviation of three replicate measurements.

Supplementary Figure 34 | The transient photocurrent density curves recorded for IAPEC system with and without sunlight irradiation in different electrolyte of 0.1 M NaCl, 0.1 M KCl, 0.05 M CaCl₂, 0.05 M MgCl₂. simulated solar light illumination 100 mW cm⁻² xenon lamp source at 30 s intervals throughout a 270 s running.

Comment 9: No error bars or standard deviations were reported for any measurements. The authors must reproduce all of their measurements three times in total with independently prepared electrodes and should report the data with averages and standard deviations. All activity should also be reported with standard errors based on appropriate error propagation using the standard deviations from the reproduced measurements. This will verify to the readers that these are reproducible results and not one-off measurements.

Response: We appreciate your valuable suggestion to provide error bars or standard deviations for our measurements. We replicated all of our measurements three times using independently prepared electrodes, as suggested. We included the averages and standard deviations in our revised manuscript for each set of measurements. Relevant revised Figures number include: **Fig.2a, Fig.4, Fig.5c, Fig. 5e, Supplementary Fig. 18, Supplementary Fig. 20-28, Supplementary Fig. 30-33, Supplementary Fig. 35-37.**

Relevant revised Figures s have been listed below:

Supplementary Figure 18 | The corresponding pseudo-first-order rate constant of IAPEC in chloride medium is as high as 0.5 min^{-1} within 60 min, while that of IAPEC in Sulfate medium PEC(0V), PEC(1V), EC and PC are $0.24, 0.12, 0.058, 0.05$ and 0.03 min^{-1} , respectively. Error bars representing the standard deviation of three replicate measurements.

Supplementary Figure 20 | Degradation of MB in different systems and concentrations (The MB concentration increased from 5 ppm to 30 ppm). Error bars representing the standard deviation of three replicate measurements.

Supplementary Figure 21 | Degradation of IBP in different systems and pollution concentrations (The IBP concentration increased from 5 ppm to 30 ppm). Error bars representing the standard deviation of three replicate measurements.

Supplementary Figure 22 | Degradation of CBZ in different systems and concentrations (The CBZ concentration increased from 1 ppm to 10 ppm). Error bars representing the standard deviation of three replicate measurements.

Supplementary Figure 23 | The degradations performances of PEC (a) and IAPEC (b) systems for Bisphenol A (BPA), 4-Chlorophenol (4-CP), Perfluorooctanoic Acid (PFOA), Sulfamethoxazole (SMX) and Cellulose Acetate Propionate (CAP). (Experimental conditions: 60 mL 0.1 M NaCl solution, initial pH = 6.0 ± 0.1, simulated solar light illumination 100 mW cm⁻², pollutant concentration: 10 ppm). Error bars representing the standard deviation of three replicate measurements.

Supplementary Figure 24 | The removal rate of TOC of eight model pollutants by IAPEC system after 2 h reaction. (Experimental conditions: 60 mL 0.1 M NaCl solution, initial pH = 6.0 ± 0.1, simulated solar light illumination 100 mW cm⁻², pollutant concentration: 10 ppm). Error bars representing the standard deviation of three replicate measurements.

Supplementary Figure 25]. Concentrations of toxic oxychlorides during the IAPEC treatment of saline sewage with different Photoanode. (N.D. is not detected, experimental conditions: 60 mL 50 mM NaCl simulated saline sewage, using TiO₂ or BiVO₄ as photoanode, conducted in triplicate). Error bars representing the standard deviation of three replicate measurements.

Revised Figure 5e] The degradation 3D line diagram of three pollutants by the IAPEC system in eight different seawater backgrounds. (Pollutant concentration: 10 ppm, the electrolytes are taken from real seawater in Liaoning, Zhejiang and Fujian). Error bars representing the standard deviation of three replicate measurements.

Supplementary Figure 35 (a) Degradation effect of different salts on MB and (b) pseudo-first-order rate constant (experimental conditions: 60 mL simulated saline sewage, initial pH = 6.0 ± 0.1, simulated solar light illumination 100 mW cm⁻², [NaCl] = 0.1 M, [Na₂SO₄] = 0.1 M, [Na₂CO₃] = 0.1 M, [NaNO₃] = 0.1 M, [NaHCO₃] = 0.1 M, [MB]₀ = 10 ppm). Error bars representing the standard deviation of three replicate measurements.

Supplementary Figure 36. The influence of the effects of humic acid (HA) on the degradation performance of the IAPEC system.. (Experimental conditions: 60 mL 0.1 M NaCl solution, initial pH = 6.0 ± 0.1, simulated solar light illumination 100 mW cm⁻²). Error bars representing the standard deviation of three replicate measurements.

Supplementary Figure 37]. The different free radical scavengers quenched. (The species and signal intensity of free radicals were determined by electron spin resonance spectroscopy (EPR). Quenching experiments were carried out with tert-butanol (TBA, $\cdot\text{OH}$, $\text{Cl}\cdot$, $\text{Cl}_2\cdot^-$, $\text{ClO}\cdot$), NaHCO_3 ($\cdot\text{OH}$, $\text{Cl}\cdot$, $\text{Cl}_2\cdot^-$), nitrobenzene (NB, $\cdot\text{OH}$), $\text{Na}_2\text{S}_2\text{O}_3$ ($\cdot\text{OH}$, $\text{Cl}\cdot$, $\text{Cl}_2\cdot^-$, $\text{ClO}\cdot$, HClO) and EDTA-2Na (h^+). Error bars representing the standard deviation of three replicate measurements.

Comment 10: The decrease in pH after the degradation of MB indicates an increase in H^+ ions (or a decrease in OH^- ions). I think the authors have to provide more evidence or reasons for these issues."

Response: Thanks for your considerable question.

In the IAPEC system during MB degradation, the photoanode generates electron-hole pairs (e^- and h^+) when exposed to light. The electrons (e^-) migrate to the PBA cathode via the external circuit, where they couple with cations. Abundant h^+ remain on the surface of photoanode and rapidly react with H_2O and Cl^- ions, producing $\cdot\text{OH}$ and $\text{Cl}\cdot$ radicals. The combination of two $\text{Cl}\cdot$ radicals can generate Cl_2 , which further reacts with H_2O to produce HClO . Subsequently, $\cdot\text{OH}$ and $\text{Cl}\cdot$ radicals react with HClO to form $\text{ClO}\cdot$ radicals with strong oxidizing capabilities. The primary reaction steps as outlined in equations (1) - (7).

We determined the concentration of free chlorine (Cl_2 , HClO/ClO) generated during the degradation of MB in IAPEC system, and found that the concentration of FCS generated under acidic conditions was increased gradually with the degradation time.

Hence, the observed decrease in pH in our system originates from the generation of H⁺ ions and free chlorine

Supplementary Figure 31]. The generation of free chlorine (FCS: Cl₂, HClO/ClO) during MB degradation in the IAPEC system. Error bars representing the standard deviation of three replicate measurements.

Based on above results and discussion, we revised the Manuscript and Supplementary Information as follows:

Line 313-315: “The decrease in pH is mainly due to the hole oxidation of H₂O and chloride ions resulting in the production of hydrogen ions and free chlorine species (FCS, Cl₂, HClO/ClO) (Supplementary Fig. 31) ³⁵.”

REVIEWERS' COMMENTS

Reviewer #1 (Remarks to the Author):

The authors have addressed all the queries in the revised manuscript. I suggest its publication in its current version.

Reviewer #2 (Remarks to the Author):

The authors have effectively addressed all the reviewer's comments, and I believe the revised manuscript is now suitable for publication. I have two minor comments:

1. In Fig. R1, the lower photocurrent observed at 4.9 cm² compared to 4 cm² appears to deviate from the current density and light-exposed area relationship as mentioned by the author.

2. In Supplementary Table 1, it is advisable to clarify that the last entry among the five samples should be renamed from 'before' to 'after' for the sake of clarity."

Point-by-point response to the reviewers' comments (in blue)

Referee #1:

General comments: The authors have addressed all the queries in the revised manuscript. I suggest its publication in its current version.

Response: We appreciate the Reviewer#1 again for his/her valuable time and efforts in reviewing our revised manuscript.

Referee #2:

General comments: The authors have effectively addressed all the reviewer's comments, and I believe the revised manuscript is now suitable for publication. I have two minor comments:

1. In Fig. R1, the lower photocurrent observed at 4.9 cm^{-2} compared to 4 cm^{-2} appears to deviate from the current density and light-exposed area relationship as mentioned by the author.

Response: We are very grateful to the reviewer for his/her meticulous comments again. We have carefully checked the original data and found that it was our mistake that wrongly matched the color of the lines with the area when drawing this figure. We have corrected our mistake in our revised manuscript.

Figure R1| The transient photocurrent values recorded for IAPEC system with and without sunlight irradiation in the electrolyte of 0.5 M NaCl, simulated solar light illumination 100 mW cm⁻² xenon lamp source at 60 s intervals.

2. In Supplementary Table 1, it is advisable to clarify that the last entry among the five samples should be renamed from 'before' to 'after' for the sake of clarity."

Response: Once again, I would like to appreciate the reviewers for his/her meticulous comments. We checked carefully and corrected our mistake.

Supplementary Table 1| Fe pre-edge peak analysis on ex situ samples and the average oxidation state of Fe species in samples.

Sample	pre-edge peak energy (eV)	E ₀ (eV)	Average oxidation state
Fe foil	-	7112	+0
FeO	7112.43	7123	+2
Fe ₂ O ₃	7115.07	7128.1	+3
Before Na ⁺ ions insertion	7114.93	7127.6	+2.9
After Na ⁺ ions insertion	7112.27	7123.4	+2.1

E₀ values obtained from the first derivation spectral, which is linearly related to the Fe valence state.